# The Price of Linear Time: Error Analysis of Structured Kernel Interpolation

**Alexander Moreno** [1]  **Justin Xiao** [2]  **Jonathan Mei** [3]

## Abstract

Structured Kernel Interpolation (SKI) (Wilson & Nickisch, 2015) scales Gaussian Processes (GPs) by approximating the kernel matrix via inducing point interpolation, achieving linear computational complexity. However, it lacks rigorous theoretical error analysis. This paper bridges this gap by proving error bounds for the SKI Gram matrix and examining their effect on hyperparameter estimation and posterior inference. We further provide a practical guide to selecting the number of inducing points under convolutional cubic interpolation: they should grow as $n^{d/3}$ for spectral norm error control. Crucially, we identify two dimensionality regimes for the SKI Gram matrix spectral norm error vs. complexity trade-off. For $d < 3$, *any* error tolerance can achieve linear time for sufficiently large sample size. For $d \geq 3$, the error must *increase* with sample size for our guarantees to hold. Our analysis provides key insights into SKI's scalability-accuracy trade-offs, establishing precise conditions for achieving linear-time GP inference with controlled error.

## 1. Introduction

Gaussian Processes (GPs) (Rasmussen & Williams, 2006) are vital stochastic processes in machine learning and statistics, with applications including spatial data analysis (Liu & Onnela, 2021), time series forecasting (Girard et al., 2002), bioinformatics (Huang et al., 2023), and Bayesian optimization (Frazier, 2018). GPs provide a non-parametric framework for modeling distributions over functions, offering flexibility and uncertainty quantification. Their ability to incorporate prior knowledge via kernel choice makes GPs effective for regression and classification.

However, GPs face substantial $O(n^3)$ (sample size $n$) computational and $O(n^2)$ memory bottlenecks, as training and inference require inverse Gram matrix computations and log-determinant calculations. These issues necessitate scalable approximations for larger datasets.

Structured Kernel Interpolation (SKI) (Wilson & Nickisch, 2015) scales GPs by approximating the kernel matrix via inducing point interpolation. For stationary kernels, it achieves $O(n + m \log m)$ ($m$ inducing points) complexity by expressing the kernel via interpolation functions and an inducing point kernel matrix. Despite its effectiveness and popularity ($> 600$ citations; (Gardner et al., 2018) has 3.5k GitHub stars), SKI lacks theoretical analysis. Key questions include: for a fixed SKI Gram matrix error using cubic convolutional interpolation, how many inducing points are needed? If $m$ is a function of $n$, when does $O(n + m \log m)$ complexity remain linear? What are the implications for hyperparameter estimation and posterior inference?

This paper bridges SKI's practice-theory gap with: 1) The first error analysis for the SKI kernel, Gram matrix (spectral norm), and related quantities, yielding guidelines for selecting inducing points ($m \propto n^{d/3}$ for spectral norm error control). 2) SKI hyperparameter estimation analysis. 3) SKI inference analysis. Key findings are: a) Two dimensionality regimes link SKI Gram matrix error and complexity: for $d < 3$, any fixed spectral error is achievable in linear time (sufficient $n$); for $d \geq 3$, the error must increase with $n$ for our linear time guarantee. b) For $\mu$-smooth log-likelihoods, SKI-based gradient ascent approaches a true stationary point's neighborhood at an $O(1/K)$ rate; neighborhood size depends on SKI score function error (ignoring responses, it scales linearly with $n$ if $m \propto n^{d/3}$). This leverages Stonyakin et al. (2023)'s results on inexact gradient descent (d'Aspremont, 2008; Devolder et al., 2014).

Sec. 2 reviews related work and Sec. 3 gives SKI background. Sec. 4 bounds key SKI errors (kernel, Gram, cross-kernel matrices). Sec. 5 analyzes SKI MLE and posterior errors. Sec. 6 presents empirical validation. Sec. 7 concludes with results, limitations, and future work.

## 2. Related Work

Related works fall into three main groups: theoretical analyses of Gaussian process regression or kernel methods using approximate kernels, SKI and its extensions, and papers

[1]MBZUAI [2]Darwin AI [3]IonQ. Correspondence to: Alexander Moreno <alexander.f.moreno@gmail.com>.

*Proceedings of the 42$^{nd}$ International Conference on Machine Learning*, Vancouver, Canada. PMLR 267, 2025. Copyright 2025 by the author(s).

| Quantity | Bound |
|---|---|
| SKI kernel error | $O(\frac{c^{2d}}{m^{3/d}})$ |
| SKI Gram matrix error | $O(\frac{nc^{2d}}{m^{3/d}})$ |
| SKI cross-kernel matrix error | $O(\frac{\max(n,T)c^{2d}}{m^{3/d}})$ |
| SKI $\frac{1}{n}$-normalized score function error | $O\left(c^{2d}\frac{\sqrt{p}\cdot n^2}{m^{3/d}}\right)$ |
| SKI posterior mean error | $\frac{c^{2d}}{\sigma^2 m^{3/d}}\sqrt{n}O\left(\max(T,n)+\frac{\sqrt{Tn}Mn}{\sigma^2}\right)$ |
| SKI posterior covariance error | $O(\frac{Tn^2mc^{4d}+\sqrt{Tn}mc^{4d}\max(T,n)}{m^{3/d}})$ |

*Table 1.* Summary of SKI error bounds with convolutional cubic interpolation. Variables: $n, T$ (train/test sizes), $d$ (dimensionality), $m$ (inducing points), $p$ (hyperparameters), $c > 0$ (constant). Key: Gram matrix error is $O(nc^{2d}m^{-3/d})$.

developing techniques we leverage. In the first group, Burt et al. (2019; 2020) analyzed the sparse variational GP framework (Titsias, 2009; Hensman et al., 2013), deriving KL divergence bounds between true and variational approximate posteriors. Moreno et al. (2023) gave bounds on SKI Gram matrix approximation error for univariate features, comparing to Nyström, but did not analyze downstream effects on approximate MLE or GP posteriors. Also, Wynne & Wild (2022); Wild et al. (2021) respectively gave a Banach space view of sparse variational GPs and Nyström connections. Finally, Modell provide entry-wise error bounds for low-rank kernel matrix approximations; our approach also uses entry-wise bounds, but theirs target the *best* low-rank approximation, while ours are for the SKI Gram matrix. Only Moreno et al. (2023) treated SKI specifically, and only in a very special case setting. Our work gives an end-to-end theory of SKI from elementwise to spectral error to estimation and posterior inference.

In the second group, Wilson & Nickisch (2015)'s foundational work, which we analyze, introduced SKI for scalable large-scale GP inference. Kapoor et al. (2021) extended SKI to high dimensions via the permutohedral lattice. Yadav et al. (2022) developed a sparse grid kernel interpolation approach to address dimensionality. Most recently, Ban et al. (2024) proposed flexible SKI adjusting grid points based on kernel hyperparameters. We focus our analysis on the original (Wilson & Nickisch, 2015) technique, though future work could extend to these latter papers' settings.

Relevant too are papers whose results or proof techniques we leverage or extend. We derive a required multivariate extension to Keys (1981)'s error analysis for convolutional cubic interpolation. We also use a recent result from the inexact gradient descent literature (Stonyakin et al., 2023) to analyze gradient ascent on the SKI log-likelihood versus the true log-likelihood. Finally, we adapt a proof technique (Bach, 2013; Musco & Musco, 2017), common for approximate kernel ridge regression in-sample error, to bound test SKI mean function error.

## 3. Gaussian Processes, Structured Kernel Interpolation and Convolutional Cubic Interpolation

This section provides background on Gaussian Processes (GPs) and two key techniques for enabling scalable inference: Structured Kernel Interpolation (SKI) and Convolutional Cubic Interpolation. SKI (Wilson & Nickisch, 2015) addresses GPs scalability issue by approximating the kernel matrix through interpolation on a set of inducing points, leveraging the efficiency of convolutional kernels. In particular, cubic convolutional kernels, as detailed in Keys (1981), provide a smooth and accurate interpolation scheme that forms the foundation of the SKI framework. In this paper, we focus on this cubic case as it is used by SKI. Future work may extend this to study higher-order interpolation methods. Here, we formally define these concepts and lay the groundwork for the subsequent error analysis.

### 3.1. Gaussian Processes

A Gaussian process $\xi \sim \text{GP}(\nu, k_\theta)$ is a stochastic process $\{\xi(\mathbf{x})\}_{\mathbf{x}\in\mathcal{X}}$ such that any finite subcollection $\{\xi(\mathbf{x}_i)\}_{i=1}^n$ is multivariate Gaussian distributed. We assume that we have index locations $\mathbf{x}_i \in \mathbb{R}^d$ and observations $y_i \in \mathbb{R}$ for a set of training points $i = 1, \ldots, n$ such that

$$y_i = \xi(\mathbf{x}_i) + \epsilon_i, \epsilon_i \sim \mathcal{N}(0, \sigma^2).$$

where $\nu : \mathcal{X} \to \mathbb{R}$, $k_\theta : \mathcal{X} \times \mathcal{X} \to \mathbb{R}$ are the prior mean and covariance functions, respectively, with $k$ a positive semi-definite (PSD) kernel with hyperparameters $\theta$. Given $\{\mathbf{x}_i, y_i\}_{i=1}^n$ or alternatively $\mathbf{X} \in \mathbb{R}^{n \times d}, \mathbf{y} \in \mathbb{R}^n$, we wish to: 1) estimate hyperparameters $\boldsymbol{\theta} \in \Theta \subseteq \mathbb{R}^p$ of kernel $k_\theta$ (e.g. RBF kernel) 2) do Bayesian inference for the posterior mean $\boldsymbol{\mu}(\cdot) \in \mathbb{R}^T$ and covariance $\boldsymbol{\Sigma}(\cdot) \in \mathbb{R}^{T \times T}$ at a set of test points $\{\mathbf{x}_t\}_{t=1}^T$. Assuming $\nu \equiv 0$ (a mean-zero GP prior), for 1), one maximizes the log-likelihood

$$\mathcal{L}(\boldsymbol{\theta}; X) = -\frac{1}{2}\mathbf{y}^\top(\mathbf{K} + \sigma^2\mathbf{I})^{-1}\mathbf{y}$$
$$-\frac{1}{2}\log|\mathbf{K} + \sigma^2\mathbf{I}| - \frac{n}{2}\log(2\pi) \quad (1)$$

to find $\boldsymbol{\theta} \in \mathcal{D} \subseteq \Theta$ where $\mathbf{K} \in \mathbb{R}^{n \times n}$ with entries $\mathbf{K}_{ij} = k_\theta(\mathbf{x}_i, \mathbf{x}_j)$ is the Gram matrix for the training dataset. For 2), given the kernel function, known observation variance $\sigma^2 > 0$ and matrix of kernel evaluations between test and training points $\mathbf{K}_{\cdot,\mathbf{X}} \in \mathbb{R}^{T \times n}$, the posterior mean and covariance are

$$\boldsymbol{\mu}(\cdot) = \mathbf{K}_{\cdot,\mathbf{X}} \left( \mathbf{K} + \sigma^2 \mathbf{I} \right)^{-1} \mathbf{y} \tag{2}$$

$$\boldsymbol{\Sigma}(\cdot) = \mathbf{K}_{\cdot,\cdot} + \sigma^2 I - \mathbf{K}_{\cdot,\mathbf{X}}(\mathbf{K} + \sigma^2 I)^{-1} \mathbf{K}_{\mathbf{X},\cdot}. \tag{3}$$

Intuitively, the GP prior represents our belief about all possible functions before seeing any data. When we observe data points, the posterior represents our updated belief - it gives higher probability to functions that fit our observations while maintaining the smoothness properties encoded in the kernel. The posterior mean can be viewed as a weighted average of these functions, where the weights depend on how well each function fits the data and satisfies the prior assumptions. The posterior variance indicates our remaining uncertainty - it is smaller near observed points where we have more confidence, and larger in regions far from our data.

A challenge is that, between the log-likelihood and the posteriors, one first needs to compute the inverse regularized Gram matrix times the response vector, $(\mathbf{K} + \sigma^2 \mathbf{I})^{-1}\mathbf{y}$. Second, one needs to compute the log-determinant $\log |\mathbf{K} + \sigma^2 \mathbf{I}|$. These are both $O(n^3)$ computationally and $O(n^2)$ memory.

### 3.2. Structured Kernel Interpolation

Structured kernel interpolation (Wilson & Nickisch, 2015) or (SKI) addresses these computational and memory bottlenecks by approximating the original kernel function $k_\theta : \mathcal{X} \times \mathcal{X} \to \mathbb{R}, \mathcal{X} \subseteq \mathbb{R}^d$ by interpolating kernel values at a chosen set of inducing points

$$\mathbf{U} = \begin{pmatrix} \mathbf{u}_1^\top \\ \vdots \\ \mathbf{u}_m^\top \end{pmatrix} \in \mathbb{R}^{m \times d}.$$

The approximate kernel $\tilde{k} : \mathcal{X} \times \mathcal{X} \to \mathbb{R}$ is then:

$$\tilde{k}(\mathbf{x}, \mathbf{x}') = \mathbf{w}(\mathbf{x})^\top \mathbf{K}_\mathbf{U} \mathbf{w}(\mathbf{x}')$$

where $\mathbf{K}_\mathbf{U} \in \mathbb{R}^{m \times m}$ is the inducing point kernel matrix, and $\mathbf{w}(\mathbf{x}), \mathbf{w}(\mathbf{x}') \in \mathbb{R}^m$ are vectors of interpolation weights using (usually cubic) convolutional kernel $u : \mathbb{R} \to \mathbb{R}$ for the points $\mathbf{x}$ and $\mathbf{x}'$, respectively. One then forms the SKI Gram matrix $\tilde{\mathbf{K}} = \mathbf{W}\mathbf{K}_\mathbf{U}\mathbf{W}^\top$ with $\mathbf{W}$ a *sparse* matrix of $L$ interpolation weights per row for a polynomial of degree $L^{1/d} - 1$. By exploiting the sparsity of each row, for stationary kernels this leads to a computational complexity of $O(nL + m \log m)$ and a memory complexity of $O(nL)$.

To learn kernel hyperparameters, one maximizes the SKI log-likelihood approximation (henceforth the SKI log-likelihood)

$$\tilde{\mathcal{L}}(\boldsymbol{\theta}; X) = -\frac{1}{2}\mathbf{y}^\top(\tilde{\mathbf{K}} + \sigma^2 \mathbf{I})^{-1}\mathbf{y}$$
$$- \frac{1}{2}\log|\tilde{\mathbf{K}} + \sigma^2 \mathbf{I}| - \frac{n}{2}\log(2\pi)$$

Given the SKI kernel $\tilde{k} : \mathcal{X} \times \mathcal{X} \to \mathbb{R}$ with learned hyperparameters, one can do posterior inference of the SKI approximations to the mean $\tilde{\boldsymbol{\mu}}(\cdot)$ and covariance $\tilde{\boldsymbol{\Sigma}}(\cdot)$ at a set of $T$ test points $\cdot$ as

$$\tilde{\boldsymbol{\mu}}(\cdot) = \tilde{\mathbf{K}}_{\cdot,\mathbf{X}} \left( \tilde{\mathbf{K}} + \sigma^2 \mathbf{I} \right)^{-1} \mathbf{y}$$
$$\tilde{\boldsymbol{\Sigma}}(\cdot) = \tilde{\mathbf{K}}_{\cdot,\cdot} + \sigma^2 I - \tilde{\mathbf{K}}_{\cdot,\mathbf{X}}(\tilde{\mathbf{K}} + \sigma^2 I)^{-1}\tilde{\mathbf{K}}_{\mathbf{X},\cdot}$$

where $\tilde{\mathbf{K}}_{\cdot,\mathbf{X}} \in \mathbb{R}^{T \times n}$ is the matrix of SKI kernels between test points and training points and $\tilde{\mathbf{K}}_{\cdot,\cdot} \in \mathbb{R}^{T \times T}$ is the SKI Gram matrix for the test points. Going forward, we may write $\mathcal{L}(\boldsymbol{\theta})$ and $\tilde{\mathcal{L}}(\boldsymbol{\theta})$, dropping the explicit dependence on the data but implying it.

### 3.3. Convolutional Cubic Interpolation

Convolutional cubic interpolation (Keys, 1981) gives a continuously differentiable interpolation of a function given its values on a regular grid, where its cubic convolutional kernel is a piecewise polynomial function that ensures continuous differentiability. We formalize this using the definitions of the cubic convolutional interpolation kernel and tensor-product cubic convolutional function below. We also define an upper bound for the sum of weights for each dimension, which will be a useful constant going forward.

**Definition 3.1.** The cubic convolutional interpolation kernel $u : \mathbb{R} \to \mathbb{R}$ is given by

$$u(s) = \begin{cases} \frac{3}{2}|s|^3 - \frac{5}{2}|s|^2 + 1, & 0 \le |s| < 1, \\ -\frac{1}{2}|s|^3 + \frac{5}{2}|s|^2 - 4|s| + 2, & 1 \le |s| < 2, \\ 0, & |s| \ge 2. \end{cases}$$

**Definition 3.2.** Let $\mathbf{x} = (x_1, x_2, ..., x_d) \in \mathbb{R}^d$ be a d-dimensional point and $f : \mathbb{R}^d \to \mathbb{R}$ a function defined on a regular grid with spacing $h$ in each dimension. Let $\mathbf{c}_\mathbf{x}$ denote the grid point closest to $\mathbf{x}$. The tensor-product cubic convolutional interpolation function $g : \mathbb{R}^d \to \mathbb{R}$ is:

$$g(\mathbf{x}) \equiv \sum_{\mathbf{k} \in \{-1,0,1,2\}^d} f(\mathbf{c}_\mathbf{x} + h\mathbf{k}) \prod_{j=1}^d u\left( \frac{x_j - (\mathbf{c}_\mathbf{x})_j - hk_j}{h} \right)$$

where $u$ is the cubic convolutional interpolation kernel and $\mathbf{k} = (k_1, \ldots, k_d)$ is a vector of integer indices.

**Definition 3.3** (Bound on interpolation weights). Let $U \subset \mathbb{R}$ be the (uniform) inducing grid with spacing $h > 0$, and let $u : \mathbb{R} \to \mathbb{R}$ be the 1-D interpolation kernel. Define

$$c \equiv \sup_{x \in \mathbb{R}} \sum_{z \in U : u\left(\frac{x-z}{h}\right) \neq 0} \left| u\left(\frac{x-z}{h}\right) \right| < \infty.$$

For cubic interpolation $u$ has compact support (only the four nearest grid points per dimension have nonzero weights), so $c < \infty$. In $d$ dimensions with tensor-product weights $\mathbf{w}(\mathbf{x})$ over $U^d$, note that each row has at most $L = 4^d$ nonzeros and

$$\|\mathbf{w}(\mathbf{x})\|_1 \leq c^d \qquad \text{for all } \mathbf{x} \in \mathbb{R}^d.$$

Going forward, we always assume that we use tensor product convolutional cubic polynomial interpolation, so that $L = 4^d$ as in (Wilson & Nickisch, 2015), but that we may vary the number of inducing points $m$. In particular, we will analyze how the number of inducing points affects error for different terms of interest, and how to choose the number of inducing points.

## 4. Important Quantities

This section derives bounds for key quantities in Structured Kernel Interpolation (SKI). Section 4.1.1 provides a bound on the elementwise error between the true kernel and its SKI approximation. In Section 4.1.2, we extend this to the spectral norm error of the SKI approximation for the training Gram matrix and train-test kernel matrix. Finally, in section 4.2 we present conditions on the number of inducing points for achieving specific error tolerance $\epsilon > 0$ and error needed to guarantee linear time complexity, noting linear time always holds for $d < 3$ with sufficiently large samples.

### 4.1. Error Bounds for the SKI Kernel

This subsection analyzes the error introduced by the SKI approximation of any *symmetric* kernel function. We start by extending the analysis of Keys (1981) to the multivariate setting, deriving error bounds for multivariate cubic convolutional polynomial interpolation. We then use these to derive the elementwise error for the SKI approximation $\tilde{k}(\mathbf{x}, \mathbf{x}')$ to a symmetric kernel, which may not be positive (semi)-definite. We next apply these elementwise bounds to derive spectral norm error bounds for SKI kernel matrices, which will be crucial for understanding the downstream effects of the SKI approximation on Gaussian process hyperparameter estimation and posterior inference.

#### 4.1.1. ELEMENTWISE

Our first lemma shows that multivariate tensor-product cubic convolutional interpolation retains error cubic in the grid spacing of Keys (1981), which is equivalent to $m^{-3/d}$ decay

with the number of inducing points $m$, but may exhibit exponential error growth with increasing dimensions.

Its proof proceeds by induction on the dimension $d$. The base case for $d = 1$ relies on the known univariate cubic convolution interpolation error bound from Keys (1981). The inductive step then demonstrates how this error bound accumulates, approximately multiplicatively with the constant $c$ for each added dimension, when extending to the tensor-product case.

**Lemma 4.1.** *The uniform error bound over a compact domain $\mathcal{X} \subseteq \mathbb{R}^d$ for tensor-product cubic convolutional interpolation of a thrice continuously differentiable function $f : \mathbb{R}^d \to \mathbb{R}$ by its interpolant $g : \mathbb{R}^d \to \mathbb{R}$ is*

$$\sup_{x \in \mathcal{X}} |f(x) - g(x)| = O(c^d h^3)$$

*or equivalently $O\left(\frac{c^d}{m^{3/d}}\right)$.*

*Proof.* See Appendix B.1.1. □

The following Lemma allows us to bound the absolute difference between the true and SKI kernels *uniformly* with the same big-$O$ error as for the underlying interpolation itself. The proof uses the the triangle inequality to decompose the error into two parts: the first is from a single interpolation, while the second from nested interpolations.

**Lemma 4.2.** *Let $\delta_{m,L}$ denote the uniform error bound of tensor-product cubic convolutional interpolation (Lemma 4.1) for $m$ inducing points and interpolation degree $L^{1/d} - 1$ (hence $L^{1/d} = 4$ for cubic interpolation, as used in SKI). The SKI kernel $\tilde{k} : \mathcal{X} \times \mathcal{X} \to \mathbb{R}$ approximating a thrice continuously differentiable (not necessarily PSD) kernel $k : \mathcal{X} \times \mathcal{X} \to \mathbb{R}$ with uniform grid spacing $h$ in each dimension has error*

$$|k(\boldsymbol{x}, \boldsymbol{x}') - \tilde{k}(\boldsymbol{x}, \boldsymbol{x}')| \leq \delta_{m,L} + c^d \delta_{m,L}$$
$$= O\left(\frac{c^{2d}}{m^{3/d}}\right).$$

*Proof.* See Appendix B.1.3 □

#### 4.1.2. SPECTRAL NORM ERROR

We now transition from elementwise error bounds to spectral norm bounds for the SKI gram matrix's approximation error, finding that it grows linearly with the sample size, exponentially with the dimension, and decays as $m^{-3/d}$ with the number of inducing points. This is not only of independent interest but also important for nearly all downstream estimation and inference analysis. We also provide a bound on the spectral norms of the SKI train/test kernel matrix's approximation error. This is useful when analyzing the GP posterior parameter error.

For this next lemma we will express it both in the general interpolation setting and again give the specific big-$O$ for convolutional cubic interpolation, but going forward we sometimes only show the latter setting in the main paper and derive the general settings in the proof. In particular, ***whenever we use big $O$-notation*** we are assuming convolutional cubic interpolation.

**Proposition 4.3.** *For the SKI approximation $\tilde{K}$ of the true Gram matrix $K$, we have*

$$\|\boldsymbol{K} - \tilde{\boldsymbol{K}}\|_2 \leq n \left( \delta_{m,L} + c^d \delta_{m,L} \right)$$
$$\equiv \gamma_{n,m,L}$$
$$= O\left( \frac{nc^{2d}}{m^{3/d}} \right)$$

*Proof.* See Appendix B.1.4 □

Its proof leverages the previous elementwise error bounds. Since the error $\mathbf{K} - \tilde{\mathbf{K}}$ is symmetric, its spectral norm is bounded by its maximum absolute row sum ($\| \cdot \|_\infty$ norm). Each row sum involves $n$ terms, each bounded by the elementwise error: this gives $n \cdot$ (elementwise error) structure. The next bound uses the property $\|\mathbf{A}\|_2 \leq \sqrt{\|\mathbf{A}\|_1 \|\mathbf{A}\|_\infty}$, which simplifies to $\|\mathbf{A}\|_2 \leq \max(\|\mathbf{A}\|_1, \|\mathbf{A}\|_\infty)$ for potentially non-square matrices. The maximum absolute column sum ($\| \cdot \|_1$) and row sum ($\| \cdot \|_\infty$) of the error matrix $K_{\cdot,X} - \tilde{K}_{\cdot,X}$ are bounded using the elementwise error from Lemma 4.2, scaled by $T$ and $n$ respectively.

**Lemma 4.4.** *Let $\boldsymbol{K}_{\cdot,X} \in \mathbb{R}^{T \times n}$ be the matrix of kernel evaluations between $T$ test points and $n$ training points, and let $\tilde{\boldsymbol{K}}_{\cdot,X} \in \mathbb{R}^{T \times n}$ be the corresponding SKI approximation. Then*

$$\|\boldsymbol{K}_{\cdot,X} - \tilde{\boldsymbol{K}}_{\cdot,X}\|_2 = O\left( \frac{\max(n,T)c^{2d}}{m^{3/d}} \right)$$

*Proof.* See Appendix B.1.5. □

### 4.2. Achieving Errors in Linear Time

Here, we show how many inducing points $m$ are sufficient to achieve a desired error tolerance $\epsilon > 0$ for the SKI Gram matrix when using cubic convolutional interpolation. Based on the Theorem, we should grow the number of inducing points at an $n^{d/3}$ rate. We then show corollaries describing 1) how $\epsilon$ and $m$ must grow to maintain linear time 2) how the dimension affects whether the error must grow with the sample size to ensure linear time SKI.

The following theorem shows the number of inducing points that will control Gram matrix error. It says that the number of inducing points should grow as $n^{d/3}$ to achieve a fixed error. The proof starts by lower bounding the desired spectral norm error with the upper bound on the actual spectral

norm error derived in Proposition 4.3: this is a sufficient condition for the desired spectral norm error to hold. It then relates the number of inducing points to the grid spacing in the SKI approximation, assuming a regular grid with equal spacing in each dimension. By substituting this relationship into the sufficient condition, the proof derives the sufficient number of inducing points to control error.

**Theorem 4.5.** *If the domain is $[-D, D]^d$, then to achieve a spectral norm error of $\|\boldsymbol{K} - \tilde{\boldsymbol{K}}\|_2 \leq \epsilon$, it is sufficient to choose the number of inducing points $m$ such that:*

$$m = \left( \frac{n}{\epsilon}(1 + c^d)K'(8c^{2d}D^3) \right)^{d/3}$$

*for some constant $K'$ that depends only on the kernel function and the interpolation scheme.*

*Proof.* See Appendix B.2.1. □

The number of inducing points should thus grow:

- Sub-linearly with the sample size and decrease in error for $d < 3$, linearly for $d = 3$ and super-linearly for $d > 3$. Thus, as we want a tighter error tolerance or have more observations we need more inducing points, but at very different rates depending on the dimensionality.

- Linearly with the volume of the domain $(2D)^d$. Thus, if our observations are concentrated in a small region and we select an appropriately sized domain to cover it we need fewer inducing points.

- Exponentially with $d^2$, as we have a $c^{2d}$ term taken to the power $d/3$. However, our empirical results suggest that this is quite pessimistic.

The next Corollary establishes a condition on the spectral norm error, $\epsilon$, that ensures linear-time $O(n)$ computational complexity for SKI. The core idea is that $\epsilon$ should be such that if we choose $m$ based on the previous Theorem, $m = O(n/\log n)$ and thus $m \log m = O(n)$.

**Corollary 4.6.** *If*

$$\epsilon \geq \frac{(1 + c^d)K'8c^{2d}D^3}{C^{3/d}} \cdot \frac{n(\log n)^{3/d}}{n^{3/d}} \tag{4}$$

*for some constants $K', C > 0$ that depend on the kernel function and the interpolation scheme and we choose $m > 0$ based on the previous theorem, then we have both $\|\boldsymbol{K} - \tilde{\boldsymbol{K}}\|_2 \leq \epsilon$ and SKI computational complexity of $O(n)$.*

*Proof.* See Appendix B.2.2. □

Interestingly, the previous Theorem and Corollary imply a fundamental difference between two dimensionality regimes. For $d < 3$, the choice of $m$ required for a fixed $\epsilon$ grows more slowly than $n/\log n$. This means that for any fixed $\epsilon > 0$, SKI with cubic interpolation is guaranteed to be a linear-time algorithm for sufficiently large $n$. In contrast, for $d \geq 3$, the choice of $m$ required for a fixed $\epsilon > 0$ eventually grows faster than $n/\log n$. Thus, to maintain linear-time complexity for $d \geq 3$ and the guarantees from Theorem 4.5, we must allow the error $\epsilon$ to increase with $n$. This demonstrates that the curse of dimensionality impacts the scalability of SKI, making it challenging to guarantee both high accuracy and linear-time complexity in higher dimensions. The next corollary formalizes this.

**Corollary 4.7.** *For $d < 3$, for any $\epsilon > 0$, Corollary 4.6 holds for any $n$ sufficiently large, so that choosing $m$ based on Theorem 4.5 is sufficient to achieve linear complexity. For $d \geq 3$, $\epsilon$ must grow with the sample size to guarantee linear complexity using Theorem 4.5.*

*Proof.* For $d < 3$, the RHS of Eqn. 4 decreases with $n$ with limit 0 and thus for sufficiently large sample size will be $\leq \epsilon$, satisfying the conditions to guarantee small error and linear time. For $d \geq 3$, the RHS of Eqn. 4 grows with $n$, so that $\epsilon$ must grow to satisfy the conditions for the guarantee. $\square$

## 5. Gaussian Processes Applications

In this section, we address how SKI affects Gaussian Processes Applications. In Section 5.1 we address how using the SKI kernel and log-likelihood affect hyperparameter estimation, showing that gradient ascent on the SKI log-likelihood approaches a ball around a stationary point of the true log-likelihood. In section 5.2 we describe how using SKI affects the accuracy of posterior inference.

### 5.1. Kernel Hyperparameter Estimation

Here we show that, for a $\mu$-smooth log-likelihood, an iterate of gradient ascent on the SKI log-likelihood approaches a neighborhood of a stationary point of the true log-likelihood at an $O\left(\frac{1}{K}\right)$ rate, with $k = 1, \ldots, K$ the iterations and with the neighborhood size determined by the SKI score function's error. To show this, we leverage a recent result for non-convex inexact gradient ascent (Stonyakin et al., 2023), which requires an upper bound on the SKI score function's error. This requires bounding the spectral norm error of the SKI Gram matrix's partial derivatives. In order to obtain this, we note that for many symmetric kernels, under weak assumptions, the partial derivatives are *also* (not necessarily PSD) symmetric kernels, and thus we can reuse the previous results directly on the partial derivatives.

Note that Stonyakin et al. (2023) does not actually imply

*convergence* to a neighborhood of a critical point, only that at least one iterate will approach it. Given the challenges of non-concave optimization and the fact that we leverage a fairly recent result, we leave stronger results to future work.

Let $\mathcal{D} \subseteq \Theta$ be a *compact* subset that we wish to optimize over. In the most precise setting we would analyze projected gradient ascent, but for simplicity we analyze gradient ascent. Let $\tilde{k}_\theta : \mathcal{X} \times \mathcal{X} \to \mathbb{R}$ be the SKI approximation of $k_\theta : \mathcal{X} \times \mathcal{X} \to \mathbb{R}$ using $m$ inducing points and interpolation degree $L - 1$. We are interested in the convergence properties of *inexact gradient ascent* using the SKI log-likelihood, e.g.

$$\boldsymbol{\theta}_{k+1} = \boldsymbol{\theta}_k + \eta \nabla \tilde{\mathcal{L}}(\boldsymbol{\theta}_k),$$

where $\eta \in \mathbb{R}$ is the learning rate and $\nabla \tilde{\mathcal{L}}(\boldsymbol{\theta}_k)$ is the SKI score function (gradient of its log-likelihood). We assume: 1) a $\mu$-smooth log-likelihood. If we optimize on a bounded domain, then for infinitely differentiable kernels (e.g. RBF) this will immediately hold. 2) that the kernel's partial derivatives are themselves symmmetric (not necessarily PSD) kernels.

**Assumption 5.1** ($\mu$-smooth-log-likelihood)**.** The true log-likelihood is $\mu$-smooth over $\mathcal{D}$. That is, for all $\boldsymbol{\theta}, \boldsymbol{\theta}' \in \mathcal{D}$,

$$\|\nabla \mathcal{L}(\boldsymbol{\theta}) - \nabla \mathcal{L}(\boldsymbol{\theta}')\| \leq \mu \|\boldsymbol{\theta} - \boldsymbol{\theta}'\|$$

**Assumption 5.2.** (Kernel Partials) For each $l \in \{1, ..., p\}$, the partial derivative of $k_\theta$ with respect to a hyperparameter $\theta_l \in \mathbb{R}$, denoted as $k'_{\theta_l}(x, x') = \frac{\partial k_\theta(x, x')}{\partial \theta_l}$, is also a symmetric (not necessarily positive semi-definite) kernel.

We next state results leading to our SKI score function error bound. We argue that we can apply the same elementwise error we derived previously to the SKI partial derivatives.

**Lemma 5.3.** *[Bound on Derivative of SKI Kernel Error using Kernel Property of Derivative] Let $\tilde{k}'_{\theta_l}(x, x')$ be the SKI approximation of $k'_{\theta_l}(x, x')$, using the same inducing points and interpolation scheme as $\tilde{k}_\theta$. Then, for all $x, x' \in \mathcal{X}$ and all $\boldsymbol{\theta} \in \Theta$, the following inequality holds:*

$$\left| \frac{\partial k_\theta(x, x')}{\partial \theta_l} - \frac{\partial \tilde{k}_\theta(x, x')}{\partial \theta_l} \right| = \left| k'_{\theta_l}(x, x') - \tilde{k}'_\theta(x, x') \right|$$

$$\leq \delta'_{m,L} + c^d \delta'_{m,L}$$

$$= O\left( \frac{c^{2d}}{m^{3/d}} \right)$$

*where $\delta'_{m,L}$ is an upper bound on the error of the SKI approximation of the kernel $k'_{\theta_l}(x, x')$ with $m$ inducing points and interpolation degree $L^{1/d} - 1$, as defined in Lemma 4.2.*

*Proof.* See Appendix C.1.1 $\square$

We then use the elementwise bound to bound the spectral norm of the SKI gram matrix's partial derivative error. This again leverages Proposition 4.3, noting these partial derivatives of the Gram matrices are themselves Gram matrices.

**Lemma 5.4.** *[Partial Derivative Gram Matrix Difference Bound] For any $l \in \{1, \ldots, p\}$,*

$$\left\|\frac{\partial \mathbf{K}}{\partial \theta_l} - \frac{\partial \tilde{\mathbf{K}}}{\partial \theta_l}\right\|_2 \leq \gamma'_{n,m,L,l}$$
$$= O\left(\frac{nc^{2d}}{m^{3/d}}\right)$$

*where $\gamma'_{n,m,L,l}$ is the bound on the spectral norm difference between the kernel matrices corresponding to $k'_{\theta_l}$ and its SKI approximation $\tilde{k}'_{\theta_l}$ (analogous to Proposition 4.3, but for the kernel $k'_{\theta_l}$).*

*Proof.* See Section C.1.2. □

We now bound the SKI score function. The key insight to the proof is that the partial derivatives of the difference between regularized gram matrix inverses is in fact a difference between two quadratic forms. We can then use standard techniques (Horn & Johnson, 2012) for bounding the difference between quadratic forms to obtain our result. The result says that, aside from the response vector's norm, the error grows quadratically in the sample size, at a square root rate in the number of hyperparameters and exponentially in the dimensionality. It further decays at an $m^{\frac{3}{d}}$ rate in the number of inducing points. Noting that to maintain Gram matrix error, $m$ should grow at an $n^{d/3}$ rate, we have that if $\|\mathbf{y}\|_2 = O(\sqrt{n})$, the $\frac{1}{n}$ normalized score function error can grow linearly with the sample size when choosing the number of inducing points based on Theorem 4.5. To control it, we actually want $m = n^{2d/3}$.

**Lemma 5.5.** *[Score Function Bound (Final Version)] Let $\nabla_\theta \mathcal{L}(\theta)$ and $\nabla_\theta \tilde{\mathcal{L}}(\theta)$ be the true and SKI log-likelihood gradients. Assume the kernel is sufficiently regular such that its partial derivatives are bounded by $\|\frac{\partial K}{\partial \theta_l}\|_2 \leq C_n$ for a constant $C_n = O(n)$.*

*Then the spectral norm of the score function error is bounded as:*

$$\|\nabla_\theta \mathcal{L}(\theta) - \nabla_\theta \tilde{\mathcal{L}}(\theta)\|_2 = O\left(c^{2d} \frac{\sqrt{p}}{m^{3/d}} \left(\|y\|_2^2 n^2 + n^3\right)\right)$$

*Furthermore, under the common assumption that $\|y\|_2^2 = O(n)$, the $\frac{1}{n}$-normalized error is bounded as:*

$$\frac{1}{n}\|\nabla_\theta \mathcal{L}(\theta) - \nabla_\theta \tilde{\mathcal{L}}(\theta)\|_2 = O\left(c^{2d} \frac{\sqrt{p} \cdot n^2}{m^{3/d}}\right)$$

*Proof.* See Section C.1.3. □

We apply Stonyakin et al. (2023) below: the result is the same as in their paper (and assumes $\mu$-smoothness as we did on $\mathcal{L}$), but using gradient ascent instead of descent and using the score function error above. It says that at an $O\left(\frac{1}{K}\right)$ rate, at least one iterate of gradient ascent has its squared gradient norm approach a neighborhood proportional to the squared SKI score function's spectral norm error.

**Theorem 5.6.** *(Stonyakin et al., 2023) For inexact gradient ascent on $\mathcal{L}$ with additively inexact gradients satisfying $\|\nabla\mathcal{L}(\boldsymbol{\theta}) - \nabla\tilde{\mathcal{L}}(\boldsymbol{\theta})\| \leq \epsilon_g$, we have:*

$$\min_{k=0,\ldots,N-1} \|\nabla\mathcal{L}(\theta_k)\|^2 \leq \frac{2\mu(\mathcal{L}^* - \mathcal{L}(\boldsymbol{\theta}_0))}{K} + \frac{\epsilon_g^2}{2\mu} \quad (5)$$

*where $\mathcal{L}^*$ is the value at a stationary point, $\mathcal{L}(\boldsymbol{\theta}_0)$ is the initial, function value, $K$ is the number of iterations and $\epsilon_g$ is the gradient error bound in the previous Lemma.*

## 5.2. Posterior Inference

Finally, we treat posterior inference. As our hyperparameter optimization results only say that *some* iterate approaches a stationary point, we will focus on the error when the SKI and true kernel hyperparameter match. We add an assumption

**Assumption 5.7.** (Bounded Kernel) Assume that the true kernel satisfies the condition that $|k(\mathbf{x}, \mathbf{x}')| \leq M$ for all $\mathbf{x}, \mathbf{x}' \in \mathcal{X}$.

Now we bound the spectral error for the SKI mean function evaluated at a set of test points. The proof follows a standard strategy commonly used for approximate kernel ridge regression. See Bach (2013); Musco & Musco (2017) for examples. The result says that the $l^2$ error may grow exponentially in the dimensionality, super-linearly in the training sample size and at worst linearly in the test sample size. It decays at an $m^{\frac{3}{d}}$ rate in the number of inducing points. For controlled error we want $m = n^{2d/3}$.

**Lemma 5.8.** *(SKI Posterior Mean Error) Let $\boldsymbol{\mu}(\cdot)$ be the GP posterior mean at a set of test points $\cdot \in \mathbb{R}^{T \times d}$ and $\tilde{\boldsymbol{\mu}}(\cdot)$ be the SKI posterior mean at those points. Then the SKI posterior mean $l^2$ error is bounded by:*

$$\|\tilde{\boldsymbol{\mu}}(\cdot) - \boldsymbol{\mu}(\cdot)\|_2$$
$$\leq \frac{c^{2d}}{\sigma^2 m^{3/d}} \sqrt{n} O\left(\max(T, n) + \frac{\sqrt{Tn}Mn}{\sigma^2}\right)$$

*Proof.* See Appendix C.2.1. □

We now derive the spectral error bound for the test SKI covariance matrix. The proof involves noticing that a key term is a difference between two quadratic forms, and using

standard techniques for bounding such a difference. The result shows that the error grows at worst super-linearly but subquadratically in the number of test points, quadratically in the training sample size and exponentially in the dimension. The error scales with the number of inducing points at an $m^{1-3/d}$ rate, so that it decays if $d < 3$. If we select the number of inducing points to be proportional to $n^{d/3}$, then the error grows at rate $n^{1+d/3}$ for $d < 3$, so that we do *not* have error control for the covariance, despite having it for the Gram matrix. A future question is whether this bound can be improved.

**Lemma 5.9.** *[SKI Posterior Covariance Error] Let $\boldsymbol{\Sigma}(\cdot)$ be the GP posterior covariance matrix at a set of test points $\cdot \in \mathbb{R}^{T \times d}$ and $\tilde{\boldsymbol{\Sigma}}(\cdot)$ be its SKI approximation. Then*

$$\|\boldsymbol{\Sigma}(\cdot) - \tilde{\boldsymbol{\Sigma}}(\cdot)\|_2$$
$$\leq \gamma_{T,m,L} + \frac{\sqrt{Tn}M}{\sigma^2} \max(\gamma_{T,m,L}, \gamma_{n,m,L})$$
$$+ \frac{\gamma_{n,m,L}}{\sigma^4} Tnmc^{2d}M^2$$
$$+ \frac{\sqrt{Tn}mc^{2d}M}{\sigma^2} \max(\gamma_{T,m,L}, \gamma_{n,m,L}).$$
$$= O\left(\frac{Tn^2mc^{4d} + \sqrt{Tn}mc^{4d}\max(T,n)}{m^{3/d}}\right).$$

*where $\gamma_{T,m,L}$ is defined as in Proposition 4.3.*

*Proof.* See Appendix C.2.2

$\square$

# 6. Empirical Analysis

To empirically investigate derived theoretical error bounds for the Structured Kernel Interpolation (SKI) Gram matrix approximation, we conducted numerical experiments. Experiments focused on spectral norm error $\|K - \tilde{K}\|_2$ behavior as a function of data points ($n$), total inducing points ($m_{total}$), and dimensionality ($d$).

## 6.1. Experimental Setup

Experiments used synthetic datasets with input points $x_i \in \mathbb{R}^d$ drawn uniformly from $[0,1]^d$. We employed a standard Radial Basis Function (RBF) kernel for all tests. SKI used cubic convolutional interpolation, consistent with our theoretical focus. All computations were performed using PyTorch and GPyTorch.

Two primary sets of experiments were run for dimensionalities $d \in \{1, 2, 3\}$:

1. **Error vs. $m_{total}$ (Fixed $n$):** For fixed $n$ (1000 for $d = 1$, 500 for $d = 2$, 250 for $d = 3$), we varied total

inducing points $m_{total}$ (with $m_{pd}$ per dimension, so $m_{total} = m_{pd}^d$) and measured $\|K - \tilde{K}\|_2$.

2. **Error vs. $n$ (Scaled $m_{total}$):** We varied $n$ (50 to 1000) and scaled $m_{total} \approx k \cdot n^{d/3}$ ($k = 1.0$), setting $m_{pd} = \max(4, \text{round}(k^{1/d}n^{1/3}))$. We then measured $\|K - \tilde{K}\|_2$.

## 6.2. Results and Discussion

The results are presented in Figure 1.

### 6.2.1. ERROR SCALING WITH NUMBER OF INDUCING POINTS ($m_{total}$)

Figure 1(a) displays spectral norm error vs. $m_{total}$ (log-log scale, fixed $n$).

- For $d = 1, 2, 3$, error shows clear power-law decay with $m_{total}$ (linear trends in log-log plot).

- Observed decay rates (slopes) are consistently steeper than the predicted $O(m_{total}^{-3/d})$: for $d = 1$, slope $\approx -5.11$ (expected $-3.00$); for $d = 2$, $\approx -2.83$ (expected $-1.50$); for $d = 3$, $\approx -2.05$ (expected $-1.00$). This suggests the $\|K - \tilde{K}\|_2 = O(nc^{2d}/m_{total}^{3/d})$ bound, while holding, may be pessimistic for smooth RBF kernels and uniform data, where actual error decay is faster.

- Error magnitude clearly increases with $d$ for fixed $m_{total}$ (e.g., at $m_{total} \approx 64$, error $d = 3 > d = 2 > d = 1$). This aligns with the $c^{2d}$ term in the bound, highlighting increased approximation challenge in higher $d$ for constant $m_{total}$.

### 6.2.2. ERROR SCALING WITH SAMPLE SIZE ($n$) AND $m_{total} \propto n^{d/3}$

Figure 1(b) shows spectral norm error vs. $n$ with $m_{total} \approx n^{d/3}$.

- For $d = 1, 2, 3$, error *decreases* with increasing $n$ when $m_{total}$ is scaled thusly—a favorable finding. Theory (substituting $m_{total}^{3/d} \propto n$ into the $O(nc^{2d}/m_{total}^{3/d})$ bound) suggests error stabilization around $O(c^{2d})$ (constant w.r.t. $n$). The observed decrease indicates better practical performance of this scaling.

- Though higher $d$ (e.g., $d = 3$) show larger error for smaller $n$, errors for $d = 1, 2, 3$ converge to similar low values as $n \to 1000$. This noteworthy convergence implies the recommended $m_{total}$ scaling effectively mitigates initial error penalties from higher $d$, enabling comparable accuracy for larger datasets.

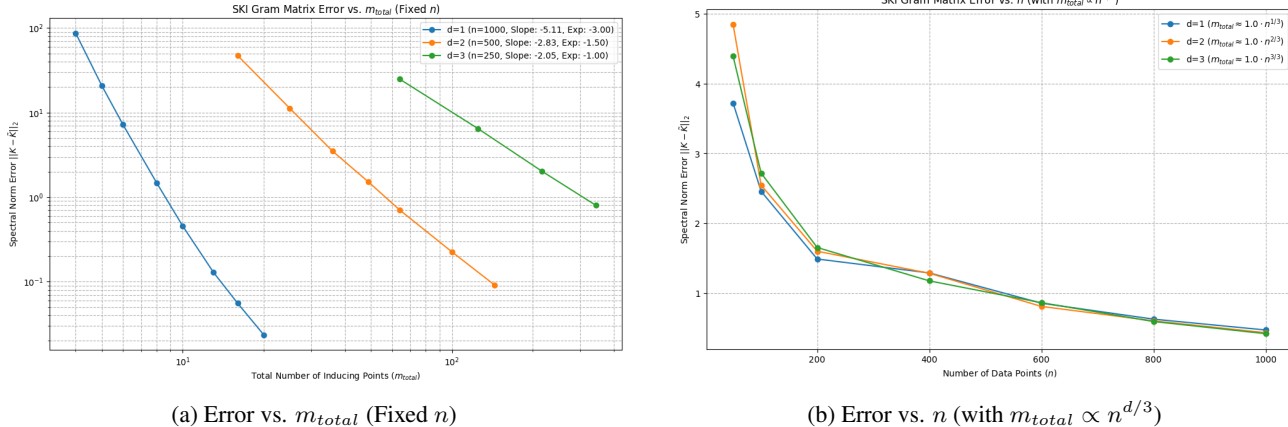

(a) Error vs. $m_{total}$ (Fixed $n$)

(b) Error vs. $n$ (with $m_{total} \propto n^{d/3}$)

*Figure 1.* Empirical SKI Gram matrix spectral norm error $||K - \tilde{K}||_2$. (a) **Error vs. $m_{total}$ (Fixed $n$):** Error decay as $m_{total}$ increases (fixed $n$: 1000 for $d = 1$, 500 for $d = 2$, 250 for $d = 3$). Log-log slopes (e.g., $\approx -5.11$ for $d = 1$, $\approx -2.83$ for $d = 2$, $\approx -2.05$ for $d = 3$) are steeper than theoretical $-3/d$ ($-3.00, -1.50, -1.00$ resp.), suggesting faster practical RBF error decay. Error magnitude increases with $d$ for given $m_{total}$. (b) **Error vs. $n$ ($m_{total} \propto n^{d/3}$):** Error as $n$ varies, with $m_{total} \approx n^{d/3}$ ($m_{pd} \approx n^{1/3}$, so $m_{total} \approx n^{d/3}$). For $d = 1, 2, 3$, error decreases as $n$ increases. Error levels for different $d$ converge for larger $n$ (e.g., $n \approx 1000$), showing scaling effectiveness against dimensionality impact.

### 6.3. Summary of Empirical Findings

Empirical results largely support our theoretical framework. SKI Gram matrix approximation error is consistent with identified dependencies on $n$, $m_{total}$, and $d$. Scaling inducing points as $m_{total} \propto n^{d/3}$ is highly effective, controlling and actively reducing approximation error as $n$ grows across tested dimensions. Observed $m_{total}$ error decay rates for RBF kernels (which are entire, the highest level of smoothness) are faster than theoretical bounds suggest, indicating conservative bounds and SKI's strong performance for smooth kernels. Error level convergence across dimensions for $m_{total} \propto n^{d/3}$ scaling at larger $n$ is particularly encouraging for practical SKI application.

## 7. Discussion

In this paper, we provided the first rigorous theoretical analysis for structured kernel interpolation. A key practical takeaway is that to control the SKI Gram matrix's spectral norm error, the number of inducing points should grow as $n^{d/3}$. Additionally, we showed the spectral norm error of the SKI gram and cross-kernel matrices, and how this impacts achieving a specific error in linear time. We then analyzed kernel hyperparameter estimation, showing that gradient ascent has an iterate approach a ball around a stationary point, where the ball's radius depends on the spectral error of the SKI score function. We showed that $m = n^{2d/3}$ sufficies to control the $\frac{1}{n}$ normalized score function error. We concluded with analysis of the error of the SKI posterior mean and variance, where $m = n^{2d/3}$ controls error for the mean function. For practitioners, the takeaway is to use SKI with $d < 3$ somewhat freely, but to use more care in the accuracy/speed tradeoff for $d \geq 3$.

This work could be extended by analyzing the error of SKI with other interpolation schemes e.g. Lagrange interpolation (Stoer et al., 1980), using potentially higher order polynomials. We could then analyze how to vary $m$ and $L$ jointly. Further, we could extend to the setting where hyperparameters (e.g. lengthscale) vary with the sample size $n$. Additionally, one could analyze the error of SKI in more complex settings, such as when the inducing points are not placed on a regular grid (Snelson & Ghahramani, 2006) or for non-stationary kernel functions, in which case the computational complexity would no longer be $O(nL+m\log m)$. Further, we analyze the optimization properties under gradient ascent: it would be interesting to analyze it under stochastic gradient ascent, analogous to Lin et al. (2024), but now using inexact noisy SKI gradients. Finally, one could analyze the methods for extending SKI to higher dimensions (Kapoor et al., 2021; Yadav et al., 2022) and for faster SKI inference (Yadav et al., 2021).

This paper used reasoning LLMs, particularly Gemini Pro 2.0. The authors contributed the paper idea and early kernel and spectral norm error analysis. LLMs helped outline the statements to be made, turn initial rough descriptions into more formal language, and make proof attempts. In general, LLM proof attempts were *wrong*, but could drive insights into a working proof strategy. We also used versions with internet access to help bring up relevant papers: hallucination rates were moderate.

## Impact Statement

This work contributes to a deeper theoretical understanding Structured Kernel Interpolation (SKI) (Wilson & Nickisch, 2015) for Gaussian Processes (GPs). By establishing error bounds and analyzing the impact of SKI on hyperparameter estimation and posterior inference, this research can lead to more confident use of approximate Gaussian Processes. These models have broad applications in various domains, including those mentioned in the introduction as well as robotics (Deisenroth et al., 2015), environmental modeling (Desai et al., 2023), and healthcare (Alaa & van der Schaar, 2017). Improved Gaussian Process models can enhance prediction accuracy and decision-making, potentially leading to advancements in robotics, more accurate environmental predictions, and better healthcare outcomes. It is important to acknowledge that the application of Gaussian Process models also carries potential risks. For instance, in healthcare, inaccurate predictions or biased models can lead to misdiagnosis or inappropriate treatment (Morley et al., 2020). Therefore, understanding potential sources of error when using approximations can be crucial to understanding how reliable we can expect them to be.

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

## A. Auxiliary Technical Results

**Lemma A.1.** *Given a function $f : \mathbb{R}^d \to \mathbb{R}$ of the form $f(x_1, x_2, ..., x_d) = \prod_{j=1}^{d} f_j(x_j)$, where each $f_j : \mathbb{R} \to \mathbb{R}$. Let $G = G^{(1)} \times G^{(2)} \times ... \times G^{(d)}$ be a fixed d-dimensional grid, where each $G^{(j)} = \{p_1^{(j)}, p_2^{(j)}, ..., p_{n_j}^{(j)}\}$ is a finite set of $n_j$ grid points along the j-th dimension for $j = 1, 2, ..., d$. Then the following equality holds:*

$$\sum_{k_1=1}^{n_1} \sum_{k_2=1}^{n_2} \cdots \sum_{k_d=1}^{n_d} \prod_{j=1}^{d} f_j(p_{k_j}^{(j)}) = \prod_{j=1}^{d} \left( \sum_{k_j=1}^{n_j} f_j(p_{k_j}^{(j)}) \right)$$

*Proof.* This is essentially a repeated application of the distributive property.

**By Induction on d (the number of dimensions):**

**Base Case (d = 1):**

When $d = 1$, the statement becomes:

$$\sum_{k_1=1}^{n_1} f_1(p_{k_1}^{(1)}) = \sum_{k_1=1}^{n_1} f_1(p_{k_1}^{(1)})$$

This is trivially true.

**Inductive Hypothesis:**

Assume the statement holds for $d = m$, i.e.,

$$\sum_{k_1=1}^{n_1} \sum_{k_2=1}^{n_2} \cdots \sum_{k_m=1}^{n_m} \prod_{j=1}^{m} f_j(p_{k_j}^{(j)}) = \prod_{j=1}^{m} \left( \sum_{k_j=1}^{n_j} f_j(p_{k_j}^{(j)}) \right)$$

**Inductive Step:**

We need to show that the statement holds for $d = m + 1$. Consider the left-hand side for $d = m + 1$:

$$\sum_{k_1=1}^{n_1} \sum_{k_2=1}^{n_2} \cdots \sum_{k_{m+1}=1}^{n_{m+1}} \prod_{j=1}^{m+1} f_j(p_{k_j}^{(j)})$$

We can rewrite this as:

$$\sum_{k_1=1}^{n_1} \sum_{k_2=1}^{n_2} \cdots \sum_{k_m=1}^{n_m} \left( \sum_{k_{m+1}=1}^{n_{m+1}} \left( \prod_{j=1}^{m} f_j(p_{k_j}^{(j)}) \right) f_{m+1}(p_{k_{m+1}}^{(m+1)}) \right)$$

Notice that the inner sum (over $k_{m+1}$) does not depend on $k_1, k_2, ..., k_m$. Thus, for any fixed values of $k_1, k_2, ..., k_m$, we can treat $\prod_{j=1}^{m} f_j(p_{k_j}^{(j)})$ as a constant. Let $C(k_1, ..., k_m) = \prod_{j=1}^{m} f_j(p_{k_j}^{(j)})$. Then we have:

$$\sum_{k_1=1}^{n_1} \sum_{k_2=1}^{n_2} \cdots \sum_{k_m=1}^{n_m} \left( C(k_1, ..., k_m) \sum_{k_{m+1}=1}^{n_{m+1}} f_{m+1}(p_{k_{m+1}}^{(m+1)}) \right)$$

Now, the inner sum $\sum_{k_{m+1}=1}^{n_{m+1}} f_{m+1}(p_{k_{m+1}}^{(m+1)})$ is a constant with respect to $k_1, ..., k_m$. Let's call this constant $S_{m+1}$. So we have:

$$\sum_{k_1=1}^{n_1} \sum_{k_2=1}^{n_2} \cdots \sum_{k_m=1}^{n_m} C(k_1, ..., k_m) S_{m+1} = S_{m+1} \sum_{k_1=1}^{n_1} \sum_{k_2=1}^{n_2} \cdots \sum_{k_m=1}^{n_m} \prod_{j=1}^{m} f_j(p_{k_j}^{(j)})$$

By the inductive hypothesis, we can replace the nested sums with a product:

$$S_{m+1} \prod_{j=1}^{m} \left( \sum_{k_j=1}^{n_j} f_j(p_{k_j}^{(j)}) \right) = \left( \sum_{k_{m+1}=1}^{n_{m+1}} f_{m+1}(p_{k_{m+1}}^{(m+1)}) \right) \prod_{j=1}^{m} \left( \sum_{k_j=1}^{n_j} f_j(p_{k_j}^{(j)}) \right)$$

Rearranging the terms, we get:

$$\prod_{j=1}^{m} \left( \sum_{k_j=1}^{n_j} f_j(p_{k_j}^{(j)}) \right) \left( \sum_{k_{m+1}=1}^{n_{m+1}} f_{m+1}(p_{k_{m+1}}^{(m+1)}) \right) = \prod_{j=1}^{m+1} \left( \sum_{k_j=1}^{n_j} f_j(p_{k_j}^{(j)}) \right)$$

This is the right-hand side of the statement for $d = m + 1$. Thus, the statement holds for $d = m + 1$.

**Conclusion:**

By induction, the statement holds for all $d \geq 1$. Therefore,

$$\sum_{k_1=1}^{n_1} \sum_{k_2=1}^{n_2} \cdots \sum_{k_d=1}^{n_d} \prod_{j=1}^{d} f_j(p_{k_j}^{(j)}) = \prod_{j=1}^{d} \left( \sum_{k_j=1}^{n_j} f_j(p_{k_j}^{(j)}) \right)$$

$\square$

*Claim* 1. Given a convex combination $\mathbf{C} = \alpha \mathbf{A} + (1 - \alpha)\mathbf{B}$, where $\alpha \in [0, 1]$, and $\mathbf{A}$ and $\mathbf{B}$ are symmetric matrices, the eigenvalues of $\mathbf{C}$ lie in the interval $[\min(\lambda_n(\mathbf{A}), \lambda_n(\mathbf{B})), \max(\lambda_1(\mathbf{A}), \lambda_1(\mathbf{B}))]$.

*Proof.* First, recall that for a symmetric matrix $\mathbf{A}$, the Rayleigh quotient $R(\mathbf{A}, \mathbf{x}) = \frac{\mathbf{x}^\top \mathbf{A} \mathbf{x}}{\mathbf{x}^\top \mathbf{x}}$ is bounded by the smallest and largest eigenvalues of $\mathbf{A}$:
$$\lambda_n(\mathbf{A}) \leq R(\mathbf{A}, \mathbf{x}) \leq \lambda_1(\mathbf{A})$$

Consider the Rayleigh quotient for the matrix $\mathbf{C}$:

$$R(\mathbf{C}, \mathbf{x}) = \frac{\mathbf{x}^\top (\alpha \mathbf{A} + (1 - \alpha)\mathbf{B})\mathbf{x}}{\mathbf{x}^\top \mathbf{x}} = \alpha R(\mathbf{A}, \mathbf{x}) + (1 - \alpha) R(\mathbf{B}, \mathbf{x})$$

Since $R(\mathbf{A}, \mathbf{x})$ and $R(\mathbf{B}, \mathbf{x})$ are bounded by their respective eigenvalues, we have:

$$R(\mathbf{C}, \mathbf{x}) \leq \alpha \lambda_1(\mathbf{A}) + (1 - \alpha) \lambda_1(\mathbf{B})$$

which implies:
$$R(\mathbf{C}, x) \leq \max(\lambda_1(\mathbf{A}), \lambda_1(\mathbf{B}))$$

Similarly,
$$R(\mathbf{C}, \mathbf{x}) \geq \min(\lambda_n(\mathbf{A}), \lambda_n(\mathbf{B}))$$

Thus, the eigenvalues of $\mathbf{C} = \alpha \mathbf{A} + (1 - \alpha)\mathbf{B}$ are bounded by:

$$\min(\lambda_n(\mathbf{A}), \lambda_n(\mathbf{B})) \leq \lambda(\mathbf{C}) \leq \max(\lambda_1(\mathbf{A}), \lambda_1(\mathbf{B}))$$

$\square$

# B. Proofs Related to Important Quantities

## B.1. Proofs Related to Ski Kernel Error Bounds

### B.1.1. PROOF OF LEMMA 4.1

**Lemma 4.1.** *The uniform error bound over a compact domain $\mathcal{X} \subseteq \mathbb{R}^d$ for tensor-product cubic convolutional interpolation of a thrice continuously differentiable function $f : \mathbb{R}^d \to \mathbb{R}$ by its interpolant $g : \mathbb{R}^d \to \mathbb{R}$ is*

$$\sup_{x \in \mathcal{X}} |f(x) - g(x)| = O(c^d h^3)$$

*or equivalently $O\left(\frac{c^d}{m^{3/d}}\right)$.*

*Proof.* We define a sequence of intermediate interpolation functions. Let $g_0(\mathbf{x}) \equiv f(\mathbf{x})$ be the original function. For $i = 1, \ldots, d$, we recursively define $g_i(\mathbf{x})$ as the function obtained by interpolating $g_{i-1}$ along the $i$-th dimension using the cubic convolution kernel $u$:

$$g_i(\mathbf{x}) \equiv \sum_{k=-1}^{2} g_{i-1}\left(\mathbf{x} + ((\mathbf{c_x})_i - x_i + kh)\, \mathbf{e}_i\right) u\left(\frac{x_i - (\mathbf{c_x})_i - kh}{h}\right).$$

Here, $\mathbf{c_x}$ is the grid point closest to $\mathbf{x}$, and $\mathbf{e}_i$ is the $i$-th standard basis vector. Thus, $g_1(\mathbf{x})$ interpolates $f$ along the first dimension, $g_2(\mathbf{x})$ interpolates $g_1$ along the second dimension, and so on, until $g_d(\mathbf{x}) = g(\mathbf{x})$ is the final tensor-product interpolated function.

We analyze the error accumulation across multiple dimensions using induction. Using (Keys, 1981), the error introduced by interpolating a thrice continuous differentiable function along a single dimension with the cubic convolution kernel is uniformly bounded over the interval domain by $Kh^3$ for some constant $K > 0$, provided the grid spacing $h$ is sufficiently small. This gives us the base case:

$$|g_1(\mathbf{x}) - g_0(\mathbf{x})| \leq Kh^3.$$

For the inductive step, assume that for some $i = k$ the error is uniformly bounded by

$$|g_k(\mathbf{x}) - g_{k-1}(\mathbf{x})| \leq c^{k-1} Kh^3.$$

We want to show that this bound also holds for $i = k + 1$. We can express the difference $g_{k+1}(\mathbf{x}) - g_k(\mathbf{x})$ as follows:

$$
\begin{aligned}
g_{k+1}(\mathbf{x}) - g_k(\mathbf{x}) &= \sum_{k_{k+1}=-1}^{2} g_k\left(\mathbf{x} + ((\mathbf{c_x})_{k+1} - x_{k+1} + k_{k+1}h)\mathbf{e}_{k+1}\right) u\left(\frac{x_{k+1} - (\mathbf{c_x})_{k+1} - k_{k+1}h}{h}\right) \\
&\quad - g_k(\mathbf{x}) \\
&= \sum_{k_{k+1}=-1}^{2}\left[\sum_{k_k=-1}^{2} g_{k-1}\left(\mathbf{x} + ((\mathbf{c_x})_k - x_k + k_kh)\mathbf{e}_k + ((\mathbf{c_x})_{k+1} - x_{k+1} + k_{k+1}h)\mathbf{e}_{k+1}\right)\right. \\
&\qquad \left. u\left(\frac{x_k - (\mathbf{c_x})_k - k_kh}{h}\right)\right] u\left(\frac{x_{k+1} - (\mathbf{c_x})_{k+1} - k_{k+1}h}{h}\right) \\
&\quad - \sum_{k_k=-1}^{2} g_{k-1}\left(\mathbf{x} + ((\mathbf{c_x})_k - x_k + k_kh)\mathbf{e}_k\right) u\left(\frac{x_k - (\mathbf{c_x})_k - k_kh}{h}\right) \\
&= \sum_{k_k=-1}^{2} u\left(\frac{x_k - (\mathbf{c_x})_k - k_kh}{h}\right) \\
&\quad \times \left[\sum_{k_{k+1}=-1}^{2} g_{k-1}\left(\mathbf{x} + ((\mathbf{c_x})_k - x_k + k_kh)\mathbf{e}_k + ((\mathbf{c_x})_{k+1} - x_{k+1} + k_{k+1}h)\mathbf{e}_{k+1}\right)\right. \\
&\qquad \left. u\left(\frac{x_{k+1} - (\mathbf{c_x})_{k+1} - k_{k+1}h}{h}\right) - g_{k-1}\left(\mathbf{x} + ((\mathbf{c_x})_k - x_k + k_kh)\mathbf{e}_k\right)\right].
\end{aligned}
$$

Denote by $I_i$ the one–dimensional cubic interpolation operator acting in coordinate $i$:

$$
(I_i h)(\mathbf{x}) \equiv \sum_{t=-1}^{2} h\left(\mathbf{x} + ((\mathbf{c_x})_i - x_i + th)\mathbf{e}_i\right) u\left(\frac{x_i - (\mathbf{c_x})_i - th}{h}\right).
$$

With this notation $g_i = I_i g_{i-1}$ and $g_0 = f$.

The inner term in the last expression is the interpolation error of $g_{k-1}$ along the $(k+1)$-st variable, evaluated at $\mathbf{y}_{k_k} := \mathbf{x} + ((\mathbf{c_x})_k - x_k + k_kh)\mathbf{e}_k$:

$$
\sum_{k_{k+1}=-1}^{2} g_{k-1}\left(\mathbf{y}_{k_k} + ((\mathbf{c_x})_{k+1} - x_{k+1} + k_{k+1}h)\mathbf{e}_{k+1}\right) u\left(\frac{x_{k+1} - (\mathbf{c_x})_{k+1} - k_{k+1}h}{h}\right) - g_{k-1}(\mathbf{y}_{k_k}) = \left((I_{k+1} - \mathrm{Id})g_{k-1}\right)(\mathbf{y}_{k_k}).
$$

Hence,

$$
\left|g_{k+1}(\mathbf{x}) - g_k(\mathbf{x})\right| \leq \sum_{k_k=-1}^{2} \left|u\left(\frac{x_k - (\mathbf{c_x})_k - k_kh}{h}\right)\right| \cdot \sup_{\mathbf{y}} \left|\left((I_{k+1} - \mathrm{Id})g_{k-1}\right)(\mathbf{y})\right|.
$$

By the definition of $c$ (uniform $\ell_1$ bound on the weights), the sum of absolute weights is $\leq c$, so

$$
\left|g_{k+1}(\mathbf{x}) - g_k(\mathbf{x})\right| \leq c \cdot \|(I_{k+1} - \mathrm{Id})g_{k-1}\|_\infty.
$$

We now apply the 1-D cubic interpolation error in the $(k+1)$-st variable to $g_{k-1}$. Since previous interpolations act on coordinates $1, \ldots, k$ only, $g_{k-1}$ remains $C^3$ in $x_{k+1}$ with

$$
\|\partial_{x_{k+1}}^3 g_{k-1}\|_\infty = \|I_1 \cdots I_{k-1}(\partial_{x_{k+1}}^3 f)\|_\infty \leq \|I_1\|_{\infty\to\infty} \cdots \|I_{k-1}\|_{\infty\to\infty} \|\partial_{x_{k+1}}^3 f\|_\infty \leq c^{k-1} \|\partial_{x_{k+1}}^3 f\|_\infty.
$$

Thus there exists a constant $K > 0$ (depending on $\|\partial_{x_{k+1}}^3 f\|_\infty$ but not on $h$) such that

$$
\|(I_{k+1} - \mathrm{Id})g_{k-1}\|_\infty \leq K c^{k-1} h^3,
$$

and therefore

$$\left|g_{k+1}(\mathbf{x}) - g_k(\mathbf{x})\right| \le c \cdot K \, c^{k-1} h^3 = c^k K h^3.$$

This completes the inductive step.

Finally, we bound the total error $|g(\mathbf{x}) - f(\mathbf{x})| = |g_d(\mathbf{x}) - g_0(\mathbf{x})|$ by summing the errors introduced at each interpolation step:

$$|g(\mathbf{x}) - f(\mathbf{x})| \le \sum_{i=1}^{d} |g_i(\mathbf{x}) - g_{i-1}(\mathbf{x})| \le \sum_{i=1}^{d} c^{i-1} K h^3 = K h^3 \sum_{i=0}^{d-1} c^i.$$

The last sum is a geometric series, which evaluates to $K h^3 \frac{1-c^d}{1-c}$. For a fixed $c > 1$ (independent of $d$), this expression is $O(c^d)$ when $d$ is large. Therefore, tensor-product cubic convolutional interpolation has $O(c^d h^3)$ error. Finally, noticing that $h = O\left(\frac{1}{m^{1/d}}\right)$ gives us the desired result. $\qquad\square$

### B.1.2. Curse of Dimensionality for Kernel Regression

The next lemma shows that when using a product kernel for $d$-dimensional kernel regression (where cubic convolutional interpolation is a special case), the sum of weights suffers from the curse of dimensionality. The proof strategy involves expressing the multi-dimensional sum as a product of sums over each individual dimension, leveraging the initial condition on the one-dimensional bound for each dimension, and taking advantage of the structure of the Cartesian grid.

**Lemma B.1.** *Let $u : \mathbb{R} \to \mathbb{R}$ be a one-dimensional kernel function with constant $c > 0$ defined as in 3.3. Let $u_d : \mathbb{R}^d \to \mathbb{R}$ be a d-dimensional product kernel defined as:*

$$u_d\left(\frac{x - x_i}{h}\right) = \prod_{j=1}^{d} u\left(\frac{x^{(j)} - x_i^{(j)}}{h}\right),$$

*where $x = (x^{(1)}, x^{(2)}, ..., x^{(d)}) \in \mathbb{R}^d$ and $x_i = (x_i^{(1)}, x_i^{(2)}, ..., x_i^{(d)}) \in \mathbb{R}^d$ are d-dimensional points. Assume the data points $\{x_i\}_{i=1}^{n}$ (n may differ from the univariate case) lie on a fixed d-dimensional grid $G = G^{(1)} \times G^{(2)} \times ... \times G^{(d)}$, where each $G^{(j)} = \{p_1^{(j)}, p_2^{(j)}, ..., p_{n_j}^{(j)}\}$ is a finite set of $n_j$ grid points along the j-th dimension for $j = 1, 2, ..., d$. Then, for any $x \in \mathbb{R}^d$, the sum of weights in the d-dimensional kernel regression is bounded by $c^d$:*

$$\sum_{i=1}^{n} \left|u_d\left(\frac{x - x_i}{h}\right)\right| \le c^d.$$

*Proof.* Let the fixed d-dimensional grid be defined by the Cartesian product of d sets of 1-dimensional grid points: $G = G^{(1)} \times G^{(2)} \times ... \times G^{(d)}$, where $G^{(j)} = \{p_1^{(j)}, p_2^{(j)}, ..., p_{n_j}^{(j)}\}$ is the set of grid points along the j-th dimension.

We start with the sum of weights in the d-dimensional case:

$$\sum_{i=1}^{n} \left|u_d\left(\frac{x - x_i}{h}\right)\right| = \sum_{i=1}^{n} \prod_{j=1}^{d} \left|u\left(\frac{x^{(j)} - x_i^{(j)}}{h}\right)\right|$$

Since the data points lie on the fixed grid $G$, we can rewrite the outer sum as a nested sum over the grid points in each dimension:

$$\sum_{i=1}^{n} \prod_{j=1}^{d} \left|u\left(\frac{x^{(j)} - x_i^{(j)}}{h}\right)\right| = \sum_{k_1=1}^{n_1} \sum_{k_2=1}^{n_2} ... \sum_{k_d=1}^{n_d} \prod_{j=1}^{d} \left|u\left(\frac{x^{(j)} - p_{k_j}^{(j)}}{h}\right)\right|$$

Now we can change the order of summation and product, as proven in Lemma A.1:

$$\sum_{k_1=1}^{n_1} \sum_{k_2=1}^{n_2} \cdots \sum_{k_d=1}^{n_d} \prod_{j=1}^{d} \left| u\left(\frac{x^{(j)} - p_{k_j}^{(j)}}{h}\right) \right| = \prod_{j=1}^{d} \left( \sum_{k_j=1}^{n_j} \left| u\left(\frac{x^{(j)} - p_{k_j}^{(j)}}{h}\right) \right| \right)$$

By the assumption of the lemma, we know that for each dimension $j$, the sum of weights is bounded by $c$. Note that $\{p_{k_j}^{(j)}\}_{k_j=1}^{n_j}$ is simply a set of points in $\mathbb{R}$, thus:

$$\sum_{k_j=1}^{n_j} \left| u\left(\frac{x^{(j)} - p_{k_j}^{(j)}}{h}\right) \right| \le c$$

Therefore, we have:

$$\prod_{j=1}^{d} \left( \sum_{k_j=1}^{n_j} \left| u\left(\frac{x^{(j)} - p_{k_j}^{(j)}}{h}\right) \right| \right) \le \prod_{j=1}^{d} c = c^d$$

Thus, we have shown that:

$$\sum_{i=1}^{n} \left| u_d\left(\frac{x - x_i}{h}\right) \right| \le c^d$$

$\square$

### B.1.3. PROOF OF LEMMA 4.2

**Lemma 4.2.** *Let $\delta_{m,L}$ denote the uniform error bound of tensor-product cubic convolutional interpolation (Lemma 4.1) for $m$ inducing points and interpolation degree $L^{1/d} - 1$ (hence $L^{1/d} = 4$ for cubic interpolation, as used in SKI). The SKI kernel $\tilde{k} : \mathcal{X} \times \mathcal{X} \to \mathbb{R}$ approximating a thrice continuously differentiable (not necessarily PSD) kernel $k : \mathcal{X} \times \mathcal{X} \to \mathbb{R}$ with uniform grid spacing $h$ in each dimension has error*

$$|k(\boldsymbol{x},\boldsymbol{x}') - \tilde{k}(\boldsymbol{x},\boldsymbol{x}')| \le \delta_{m,L} + c^d \delta_{m,L}$$
$$= O\left(\frac{c^{2d}}{m^{3/d}}\right).$$

*Proof.* Recall that SKI approximates the kernel as

$$k(\mathbf{x},\mathbf{x}') \approx \tilde{k}(\mathbf{x},\mathbf{x}')$$
$$= \boldsymbol{w}(\mathbf{x})^\top \mathbf{K_U} \boldsymbol{w}(\mathbf{x}'),$$

Let $\mathbf{K_{U,x'}} \in \mathbb{R}^m$ be the vector of kernels between the inducing points and the vector $\mathbf{x}'$

$$
\begin{aligned}
|k(\mathbf{x},\mathbf{x}') - \tilde{k}(\mathbf{x},\mathbf{x}')| &= |k(\mathbf{x},\mathbf{x}') - \boldsymbol{w}(\mathbf{x})^\top \mathbf{K_{U,x'}} + \boldsymbol{w}(\mathbf{x})^\top \mathbf{K_{U,x'}} - \boldsymbol{w}(\mathbf{x})^\top \mathbf{K_U} \boldsymbol{w}(\mathbf{x}')| \\
&\le |k(\mathbf{x},\mathbf{x}') - \boldsymbol{w}(\mathbf{x})^\top \mathbf{K_{U,x'}}| + |\boldsymbol{w}(\mathbf{x})^\top \mathbf{K_{U,x'}} - \boldsymbol{w}(\mathbf{x})^\top \mathbf{K_U} \boldsymbol{w}(\mathbf{x}')| \\
&\le \delta_{m,L} + |\boldsymbol{w}(\mathbf{x})^\top \mathbf{K_{U,x'}} - \boldsymbol{w}(\mathbf{x})^\top \mathbf{K_U} \boldsymbol{w}(\mathbf{x}')| \\
&\quad \text{since } |k(\mathbf{x},\mathbf{x}') - \boldsymbol{w}(\mathbf{x})^\top \mathbf{K_{U,x'}}| \text{ is a single polynomial interpolation}
\end{aligned}
\tag{6}
$$

Now note that $\mathbf{w}(x) \in \mathbb{R}^m$ is a sparse matrix with at most $L$ non-zero entries. Thus, letting $\tilde{\mathbf{w}}(x) \in \mathbb{R}^L$ be the non-zero entries of $\mathbf{w}(x)$ and similarly $\tilde{\mathbf{K}}_{\mathbf{U},\mathbf{x}'} \in \mathbb{R}^L$ be the entries of $\mathbf{K}_{\mathbf{U},\mathbf{x}'}$ in the dimensions corresponding to non-zero entries of

$\mathbf{w}(x) \in \mathbb{R}^m$, while $\tilde{\mathbf{K}}_{\mathbf{U}} \in \mathbb{R}^{L \times m}$ is the analogous matrix for $\mathbf{K}_{\mathbf{U}}$, we have

$$
\begin{aligned}
|\boldsymbol{w}(\mathbf{x})^\top \mathbf{K}_{\mathbf{U},\mathbf{x}'} - \boldsymbol{w}(\mathbf{x})^\top \mathbf{K}_{\mathbf{U}} \boldsymbol{w}(\mathbf{x}')| &= |\tilde{\mathbf{w}}(\mathbf{x})^\top \tilde{\mathbf{K}}_{\mathbf{U},\mathbf{x}'} - \tilde{\mathbf{w}}(\mathbf{x})^\top \tilde{\mathbf{K}}_{\mathbf{U}} \boldsymbol{w}(\mathbf{x}')| \\
&\leq \|\tilde{\mathbf{w}}(\mathbf{x})\|_1 \|\tilde{\mathbf{K}}_{\mathbf{U},\mathbf{x}'} - \tilde{\mathbf{K}}_{\mathbf{U}} \boldsymbol{w}(\mathbf{x}')\|_\infty \\
&\leq c^d \|\tilde{\mathbf{K}}_{\mathbf{U},\mathbf{x}'} - \tilde{\mathbf{K}}_{\mathbf{U}} \boldsymbol{w}(\mathbf{x}')\|_\infty \quad \text{Lemma B.1} \\
&\leq c^d \delta_{m,L}
\end{aligned}
\tag{7}
$$

where the last line follows as each element of $\mathbf{K}_{\mathbf{U}} \boldsymbol{w}(\mathbf{x}')$ is a polynomial interpolation approximating each element of $\mathbf{K}_{\mathbf{U},\mathbf{x}'}$. Plugging Eqn. 7 into Eqn. 6 gives us the desired initial result of

$$
|k(\mathbf{x}, \mathbf{x}') - \tilde{k}(\mathbf{x}, \mathbf{x}')| \leq \delta_{m,L} + c^d \delta_{m,L}
$$

and Lemma 4.1 gives us the result when the convolutional kernel is cubic. $\qquad \square$

### B.1.4. PROOF OF PROPOSITION 4.3

**Proposition 4.3.** *For the SKI approximation $\tilde{K}$ of the true Gram matrix $K$, we have*

$$
\begin{aligned}
\|\boldsymbol{K} - \tilde{\boldsymbol{K}}\|_2 &\leq n \left( \delta_{m,L} + c^d \delta_{m,L} \right) \\
&\equiv \gamma_{n,m,L} \\
&= O \left( \frac{n c^{2d}}{m^{3/d}} \right)
\end{aligned}
$$

*Proof.* Recall that for any matrix $\mathbf{A}$, $\|\mathbf{A}\|_2 \leq \sqrt{\|\mathbf{A}\|_1 \|\mathbf{A}\|_\infty}$. Since $\mathbf{K} - \tilde{\mathbf{K}}$ is symmetric, we have

$$
\|\mathbf{K} - \tilde{\mathbf{K}}\|_2 \leq \sqrt{\|\mathbf{K} - \tilde{\mathbf{K}}\|_1 \|\mathbf{K} - \tilde{\mathbf{K}}\|_\infty} = \|\mathbf{K} - \tilde{\mathbf{K}}\|_\infty
$$

Furthermore, $\|\mathbf{K} - \tilde{\mathbf{K}}\|_\infty$ is the maximum absolute row sum of $\mathbf{K} - \tilde{\mathbf{K}}$. Since there are $n$ rows and, by Lemma 4.2, each element of $\mathbf{K} - \tilde{\mathbf{K}}$ is bounded by $\delta_{m,L} + c^d \delta_{m,L}$ in absolute value, we have

$$
\|\mathbf{K} - \tilde{\mathbf{K}}\|_\infty \leq n \left( \delta_{m,L} + c^d \delta_{m,L} \right) = \gamma_{n,m,L}.
$$

Therefore, $\|\mathbf{K} - \tilde{\mathbf{K}}\|_2 \leq \gamma_{n,m,L}$. $\qquad \square$

### B.1.5. PROOF OF LEMMA 4.4

**Lemma 4.4.** *Let $K_{\cdot,X} \in \mathbb{R}^{T \times n}$ be the matrix of kernel evaluations between $T$ test points and $n$ training points, and let $\tilde{K}_{\cdot,X} \in \mathbb{R}^{T \times n}$ be the corresponding SKI approximation. Then*

$$
\|K_{\cdot,X} - \tilde{K}_{\cdot,X}\|_2 = O \left( \frac{\max(n, T) c^{2d}}{m^{3/d}} \right)
$$

*Proof.* Using the same reasoning as in Proposition 4.3, we have

$$
\begin{aligned}
\|\mathbf{K}_{\cdot,\mathbf{X}} - \tilde{\mathbf{K}}_{\cdot,\mathbf{X}}\|_2 &\leq \sqrt{\|\mathbf{K}_{\cdot,\mathbf{X}} - \tilde{\mathbf{K}}_{\cdot,\mathbf{X}}\|_1 \|\mathbf{K}_{\cdot,\mathbf{X}} - \tilde{\mathbf{K}}_{\cdot,\mathbf{X}}\|_\infty} \\
&\leq \max \left( \|\mathbf{K}_{\cdot,\mathbf{X}} - \tilde{\mathbf{K}}_{\cdot,\mathbf{X}}\|_1, \|\mathbf{K}_{\cdot,\mathbf{X}} - \tilde{\mathbf{K}}_{\cdot,\mathbf{X}}\|_\infty \right).
\end{aligned}
$$

Now, $\|\mathbf{K}_{\cdot,\mathbf{X}} - \tilde{\mathbf{K}}_{\cdot,\mathbf{X}}\|_1$ is the maximum absolute column sum, which is less than or equal to $T(\delta_{m,L} + c^d \delta_{m,L}) = \gamma_{T,m,L}$. Similarly, $\|\mathbf{K}_{\cdot,\mathbf{X}} - \tilde{\mathbf{K}}_{\cdot,\mathbf{X}}\|_\infty$ is the maximum absolute row sum, which is upper bounded by $n(\delta_{m,L} + c^d \delta_{m,L}) = \gamma_{n,m,L}$. Therefore,

$$
\|\mathbf{K}_{\cdot,\mathbf{X}} - \tilde{\mathbf{K}}_{\cdot,\mathbf{X}}\|_2 \leq \max(\gamma_{T,m,L}, \gamma_{n,m,L}).
$$

$\qquad \square$

B.1.6. ADDITIONAL SPECTRAL NORM BOUNDS

**Lemma B.2.** *Let $\boldsymbol{K}_{\cdot,\boldsymbol{X}} \in \mathbb{R}^{T \times n}$ be cross kernel matrix between $T$ test points and $n$ training points, where the SKI approximation uses $m$ inducing points. If the kernel function $k$ is bounded such that $|k(\boldsymbol{x}, \boldsymbol{x}')| \leq M$ for all $\boldsymbol{x}, \boldsymbol{x}' \in \mathcal{X}$, then:*

$$\|\boldsymbol{K}_{\cdot,\boldsymbol{X}}\|_2 \leq \sqrt{Tn}M$$

*Proof.*

$$\|\mathbf{K}_{\cdot,\mathbf{X}}\|_2 \leq \sqrt{\|\mathbf{K}_{\cdot,\mathbf{X}}\|_1 \|\mathbf{K}_{\cdot,\mathbf{X}}\|_\infty}$$
$$\leq \sqrt{Tn}M$$

$\square$

**Lemma B.3.** *Let $\tilde{\boldsymbol{K}}_{\cdot,\boldsymbol{X}} \in \mathbb{R}^{T \times n}$ be the matrix of SKI kernel evaluations between $T$ test points and $n$ training points, where the SKI approximation uses $m$ inducing points. Let $\boldsymbol{W}(\cdot) \in \mathbb{R}^{T \times m}$ and $\boldsymbol{W}(\boldsymbol{X}) \in \mathbb{R}^{n \times m}$ be the matrices of interpolation weights for the test points and training points, respectively. Assume that the interpolation scheme is such that the sum of the absolute values of the interpolation weights for any point is bounded by $c^d$, where $c > 0$ is a constant. Let $\boldsymbol{K}_U \in \mathbb{R}^{m \times m}$ be the kernel matrix evaluated at the inducing points. If the kernel function $k$ is bounded such that $|k(\boldsymbol{x}, \boldsymbol{x}')| \leq M$ for all $\boldsymbol{x}, \boldsymbol{x}' \in \mathcal{X}$, then:*

$$\|\tilde{\boldsymbol{K}}_{\cdot,\boldsymbol{X}}\|_2 \leq \sqrt{Tn}mc^{2d}M$$

*Proof.* By the definition of the SKI approximation and the submultiplicativity of the spectral norm, we have:

$$\|\tilde{\mathbf{K}}_{\cdot,\mathbf{X}}\|_2 = \|\mathbf{W}(\cdot)\mathbf{K}_{\mathbf{U}}(\mathbf{W}(\mathbf{X}))^\top\|_2 \leq \|\mathbf{W}(\cdot)\|_2 \|\mathbf{K}_{\mathbf{U}}\|_2 \|\mathbf{W}(\mathbf{X})\|_2$$

We now bound each term.

1. Bounding $\|\mathbf{W}(\cdot)\|_2$ and $\|\mathbf{W}(\mathbf{X})\|_2$: Since the spectral norm is induced by the Euclidean norm, and using the assumption that the sum of absolute values of interpolation weights for any point is bounded by $c^d$, we have

$$\|\mathbf{W}(\cdot)\|_2 \leq \sqrt{\|\mathbf{W}(\cdot)\|_1 \|\mathbf{W}(\cdot)\|_\infty} \leq \sqrt{Tc^d \cdot c^d} = \sqrt{T}c^d.$$

Similarly, $\|\mathbf{W}(\mathbf{X})\|_2 \leq \sqrt{n}c^d$.

2. Bounding $\|\mathbf{K}_{\mathbf{U}}\|_2$: Since $\mathbf{K}_{\mathbf{U}}$ is symmetric, $\|\mathbf{K}_{\mathbf{U}}\|_2 \leq \|\mathbf{K}_{\mathbf{U}}\|_\infty$. Each entry of $\mathbf{K}_{\mathbf{U}}$ is bounded by $M$ (by the boundedness of $k$), and each row has $m$ entries, so $\|\mathbf{K}_{\mathbf{U}}\|_\infty \leq mM$. Thus, $\|\mathbf{K}_{\mathbf{U}}\|_2 \leq mM$.

Combining these bounds, we get:

$$\|\tilde{\mathbf{K}}_{\cdot,\mathbf{X}}\|_2 \leq (\sqrt{T}c^d)(mM)(\sqrt{n}c^d) = \sqrt{Tn}mc^{2d}M$$

as required.
$\square$

**Lemma B.4.** *Let $\tilde{\mathbf{K}}$ be the SKI approximation of the kernel matrix $\mathbf{K}$, and $\sigma^2 > 0$ be the observation variance. Assume that $\boldsymbol{K}, \tilde{\boldsymbol{K}}$ are PSD. The spectral error of the regularized inverse can be bounded as follows:*

$$\left\| \left(\tilde{\mathbf{K}} + \sigma^2\mathbf{I}\right)^{-1} - \left(\mathbf{K} + \sigma^2\mathbf{I}\right)^{-1} \right\|_2 \leq \frac{\gamma_{n,m,L}}{\sigma^4}$$

*Proof.* Note that

$$\left(\tilde{\mathbf{K}} + \sigma^2\mathbf{I}\right)^{-1} - \left(\mathbf{K} + \sigma^2\mathbf{I}\right)^{-1} = \left(\tilde{\mathbf{K}} + \sigma^2\mathbf{I}\right)^{-1}\left(\mathbf{K} - \tilde{\mathbf{K}}\right)\left(\mathbf{K} + \sigma^2\mathbf{I}\right)^{-1}$$

Taking the spectral norm, we have

$$\left\| \left(\tilde{\mathbf{K}} + \sigma^2 \mathbf{I}\right)^{-1} - \left(\mathbf{K} + \sigma^2 \mathbf{I}\right)^{-1} \right\|_2 \leq \left\| \left(\tilde{\mathbf{K}} + \sigma^2 \mathbf{I}\right)^{-1} \right\|_2 \|\mathbf{K} - \tilde{\mathbf{K}}\|_2 \left\| \left(\mathbf{K} + \sigma^2 \mathbf{I}\right)^{-1} \right\|_2$$

$$\leq \gamma_{n,m,L} \left\| \left(\tilde{\mathbf{K}} + \sigma^2 \mathbf{I}\right)^{-1} \right\|_2 \left\| \left(\mathbf{K} + \sigma^2 \mathbf{I}\right)^{-1} \right\|_2 \text{ by Proposition 4.3}$$

$$\leq \frac{\gamma_{n,m,L}}{\sigma^4}$$

$\square$

## B.2. Proofs Related to Linear Time Analysis

### B.2.1. PROOF OF THEOREM 4.5

**Theorem 4.5.** *If the domain is $[-D, D]^d$, then to achieve a spectral norm error of $\|\mathbf{K} - \tilde{\mathbf{K}}\|_2 \leq \epsilon$, it is sufficient to choose the number of inducing points $m$ such that:*

$$m = \left( \frac{n}{\epsilon}(1 + c^d)K'(8c^{2d}D^3) \right)^{d/3}$$

*for some constant $K'$ that depends only on the kernel function and the interpolation scheme.*

*Proof.* We want to choose $m$ such that the spectral norm error $\|\mathbf{K} - \tilde{\mathbf{K}}\|_2 \leq \epsilon$. From Proposition 4.3, we have the bound:

$$\|\mathbf{K} - \tilde{\mathbf{K}}\|_2 \leq n(1 + c^d)\delta_{m,L}$$

For cubic interpolation, from Lemma 4.1, we have the scaling for the elementwise error:

$$\delta_{m,L} \leq K'c^{2d}h^3$$

where $K'$ is a constant that depends only on the kernel function and the interpolation scheme. Therefore, a sufficient condition to ensure $\|\mathbf{K} - \tilde{\mathbf{K}}\|_2 \leq \epsilon$ is:

$$n(1 + c^d)K'c^{2d}h^3 \leq \epsilon \tag{8}$$

The inducing points are placed on a regular grid with spacing $h$ in each dimension over the domain $[-D, D]^d$. The number of inducing points $m$ is related to $h$ by:

$$h = \frac{2D}{m^{1/d}}$$

Substituting this into the sufficient condition (8), we get:

$$n(1 + c^d)K'c^{2d} \left( \frac{2D}{m^{1/d}} \right)^3 \leq \epsilon$$

Rearranging to isolate $m$, we obtain:

$$m^{3/d} \geq \frac{n}{\epsilon}(1 + c^d)K'c^{2d}(8D^3)$$

$$m \geq \left( \frac{n}{\epsilon}(1 + c^d)K'(8c^{2d}D^3) \right)^{d/3}$$

$\square$

### B.2.2. PROOF OF COROLLARY 4.6

**Corollary 4.6.** *If*

$$\epsilon \geq \frac{(1 + c^d)K'8c^{2d}D^3}{C^{3/d}} \cdot \frac{n(\log n)^{3/d}}{n^{3/d}} \tag{4}$$

*for some constants $K', C > 0$ that depend on the kernel function and the interpolation scheme and we choose $m > 0$ based on the previous theorem, then we have both $\|\boldsymbol{K} - \tilde{\boldsymbol{K}}\|_2 \leq \epsilon$ and SKI computational complexity of $O(n)$.*

*Proof.* Assume that

$$\epsilon \geq \frac{(1 + 2c^d)K'8c^{2d}D^3}{C^{3/d}} \cdot \frac{n(\log n)^{3/d}}{n^{3/d}}.$$

Rearranging this we obtain

$$\left(\frac{n}{\epsilon}(1 + 2c^d)K'(8c^{2d}D^3)\right)^{d/3} \leq C\frac{n}{\log n}.$$

$$= O\left(\frac{n}{\log n}\right).$$

Now taking

$$m = \left(\frac{n}{\epsilon}(1 + c^d)K'(8c^{2d}D^3)\right)^{d/3}$$

we have that $m = O\left(\frac{n}{\log n}\right)$ and by Theorem 4.5, $\|\mathbf{K} - \tilde{\mathbf{K}}\|_2 \leq \epsilon$. Now plugging in $\frac{n}{\log n}$ into $m \log m$ we obtain

$$O\left(m \log m\right) = O\left(\frac{n}{\log n} \log \frac{n}{\log n}\right)$$

$$= O\left(\frac{n}{\log n} \log n - \frac{n}{\log n} \log \log n\right)$$

$$= O(n)$$

as desired. $\qquad\square$

## C. Proofs Related to Gaussian Process Applications

### C.1. Proofs Related to Hyperparameter Estimation

### C.1.1. PROOF OF LEMMA 5.3

**Lemma 5.3.** *[Bound on Derivative of SKI Kernel Error using Kernel Property of Derivative] Let $\tilde{k}'_{\theta_l}(x, x')$ be the SKI approximation of $k'_{\theta_l}(x, x')$, using the same inducing points and interpolation scheme as $\tilde{k}_\theta$. Then, for all $x, x' \in \mathcal{X}$ and all $\boldsymbol{\theta} \in \Theta$, the following inequality holds:*

$$\left|\frac{\partial k_\theta(x, x')}{\partial \theta_l} - \frac{\partial \tilde{k}_\theta(x, x')}{\partial \theta_l}\right| = \left|k'_{\theta_l}(x, x') - \tilde{k}'_\theta(x, x')\right|$$

$$\leq \delta'_{m,L} + c^d \delta'_{m,L}$$

$$= O\left(\frac{c^{2d}}{m^{3/d}}\right)$$

*where $\delta'_{m,L}$ is an upper bound on the error of the SKI approximation of the kernel $k'_{\theta_l}(x, x')$ with $m$ inducing points and interpolation degree $L^{1/d} - 1$, as defined in Lemma 4.2.*

*Proof.* By assumption, $k'_{\theta_i}(x, x') = \frac{\partial k_\theta(x,x')}{\partial \theta_i}$ is a symmetric kernel. The SKI approximation of $k'_{\theta_i}(x, x')$ using the same inducing points and interpolation scheme as $\tilde{k}_\theta(x, x')$ is given by $\tilde{k}'_\theta(x, x')$. For the kernel $k'_{\theta_i}(x, x')$, we have:

$$\left| k'_{\theta_i}(x, x') - \tilde{k}'_\theta(x, x') \right| \leq \delta'_{m,L},$$

where $\delta'_{m,L}$ is the upper bound on the error of the SKI approximation of $k'_{\theta_i}(x, x')$ as defined in Lemma 4.2.

Now, we need to show that $\frac{\partial \tilde{k}_\theta(x,x')}{\partial \theta_i} = \tilde{k}'_\theta(x, x')$. Recall that the SKI approximation $\tilde{k}_\theta(x, x')$ is a linear combination of kernel evaluations at inducing points, with weights that depend on $x$ and $x'$:

$$\tilde{k}_\theta(x, x') = \sum_{j=1}^{m} \sum_{l=1}^{m} w_{jl}(x, x') k_\theta(u_j, u_l)$$

where $w_{jl}(x, x')$ are the interpolation weights. Taking the partial derivative with respect to $\theta_i$, we get:

$$\begin{aligned}
\frac{\partial \tilde{k}_\theta(x, x')}{\partial \theta_i} &= \sum_{j=1}^{m} \sum_{l=1}^{m} w_{jl}(x, x') \frac{\partial k_\theta(u_j, u_l)}{\partial \theta_i} \\
&= \sum_{j=1}^{m} \sum_{l=1}^{m} w_{jl}(x, x') k'_{\theta_i}(u_j, u_l).
\end{aligned}$$

This is precisely the SKI approximation of the kernel $k'_{\theta_i}(x, x')$ using the same inducing points and weights:

$$\tilde{k}'_\theta(x, x') = \sum_{j=1}^{m} \sum_{l=1}^{m} w_{jl}(x, x') k'_{\theta_i}(u_j, u_l).$$

Therefore, $\frac{\partial \tilde{k}_\theta(x,x')}{\partial \theta_i} = \tilde{k}'_\theta(x, x')$.

Substituting this into our inequality, we get:

$$\begin{aligned}
\left| \frac{\partial k_\theta(x, x')}{\partial \theta_i} - \frac{\partial \tilde{k}_\theta(x, x')}{\partial \theta_i} \right| &= \left| k'_{\theta_i}(x, x') - \tilde{k}'_\theta(x, x') \right| \\
&\leq \delta'_{m,L} + c^d \delta'_{m,L}.
\end{aligned}$$

$\square$

### C.1.2. PROOF OF LEMMA 5.4

**Lemma 5.4.** *[Partial Derivative Gram Matrix Difference Bound] For any $l \in \{1, \ldots, p\}$,*

$$\begin{aligned}
\left\| \frac{\partial \mathbf{K}}{\partial \theta_l} - \frac{\partial \tilde{\mathbf{K}}}{\partial \theta_l} \right\|_2 &\leq \gamma'_{n,m,L,l} \\
&= O\left( \frac{nc^{2d}}{m^{3/d}} \right)
\end{aligned}$$

*where $\gamma'_{n,m,L,l}$ is the bound on the spectral norm difference between the kernel matrices corresponding to $k'_{\theta_l}$ and its SKI approximation $\tilde{k}'_{\theta_l}$ (analogous to Proposition 4.3, but for the kernel $k'_{\theta_l}$).*

*Proof.* Let $K'_{\theta,l}$ be the kernel matrix corresponding to the kernel $k'_{\theta,l}(x, x') = \frac{\partial k_\theta(x,x')}{\partial \theta_l}$, and let $\tilde{K}'_{\theta,l}$ be the kernel matrix corresponding to its SKI approximation $\tilde{k}'_{\theta,l}(x, x')$.

From Lemma 5.3, we have:

$$\frac{\partial \tilde{k}_\theta(x, x')}{\partial \theta_l} = \tilde{k}'_{\theta,l}(x, x') \tag{9}$$

Therefore:

$$\frac{\partial K}{\partial \theta_l} - \frac{\partial \tilde{K}}{\partial \theta_l} = K'_{\theta,l} - \tilde{K}'_{\theta,l} \tag{10}$$

By Proposition 4.3, we have a bound on the spectral norm difference between a kernel matrix and its SKI approximation. Let $\gamma'_{n,m,L,l}$ be the corresponding bound for the kernel $k'_{\theta,l}$ and its SKI approximation $\tilde{k}'_{\theta,l}$. Then:

$$\|K'_{\theta,l} - \tilde{K}'_{\theta,l}\|_2 \leq \gamma'_{n,m,L,l} \tag{11}$$

Thus,

$$\left\|\frac{\partial K}{\partial \theta_l} - \frac{\partial \tilde{K}}{\partial \theta_l}\right\|_2 = \|K'_{\theta,l} - \tilde{K}'_{\theta,l}\|_2 \leq \gamma'_{n,m,L,l}$$

This completes the proof. $\square$

### C.1.3. PROOF OF LEMMA 5.5

**Lemma 5.5.** *[Score Function Bound (Final Version)] Let $\nabla_\theta \mathcal{L}(\theta)$ and $\nabla_\theta \tilde{\mathcal{L}}(\theta)$ be the true and SKI log-likelihood gradients. Assume the kernel is sufficiently regular such that its partial derivatives are bounded by $\|\frac{\partial K}{\partial \theta_l}\|_2 \leq C_n$ for a constant $C_n = O(n)$.*

*Then the spectral norm of the score function error is bounded as:*

$$\|\nabla_\theta \mathcal{L}(\theta) - \nabla_\theta \tilde{\mathcal{L}}(\theta)\|_2 = O\left(c^{2d}\frac{\sqrt{p}}{m^{3/d}}\left(\|y\|_2^2 n^2 + n^3\right)\right)$$

*Furthermore, under the common assumption that $\|y\|_2^2 = O(n)$, the $\frac{1}{n}$-normalized error is bounded as:*

$$\frac{1}{n}\|\nabla_\theta \mathcal{L}(\theta) - \nabla_\theta \tilde{\mathcal{L}}(\theta)\|_2 = O\left(c^{2d}\frac{\sqrt{p} \cdot n^2}{m^{3/d}}\right)$$

*Proof.* We start with the expressions for the gradients:

$$\nabla \mathcal{L}(\theta) = \nabla\left(-\frac{1}{2}\mathbf{y}^\top(\mathbf{K} + \sigma^2\mathbf{I})^{-1}\mathbf{y} - \frac{1}{2}\log|\mathbf{K} + \sigma^2\mathbf{I}| - \frac{n}{2}\log(2\pi)\right).$$

$$\nabla \tilde{\mathcal{L}}(\theta) = \nabla\left(-\frac{1}{2}\mathbf{y}^\top(\tilde{\mathbf{K}} + \sigma^2\mathbf{I})^{-1}\mathbf{y} - \frac{1}{2}\log|\tilde{\mathbf{K}} + \sigma^2\mathbf{I}| - \frac{n}{2}\log(2\pi)\right).$$

Thus, the difference is:

$$\|\nabla \mathcal{L}(\theta) - \nabla \tilde{\mathcal{L}}(\theta)\|_2 = \left\|\nabla \left(-\frac{1}{2}\mathbf{y}^\top (\mathbf{K} + \sigma^2 \mathbf{I})^{-1}\mathbf{y} - \frac{1}{2}\log|\mathbf{K} + \sigma^2 \mathbf{I}|\right)\right.$$

$$\left.-\nabla \left(-\frac{1}{2}\mathbf{y}^\top (\tilde{\mathbf{K}} + \sigma^2 \mathbf{I})^{-1}\mathbf{y} - \frac{1}{2}\log|\tilde{\mathbf{K}} + \sigma^2 \mathbf{I}|\right)\right\|_2$$

$$\leq \underbrace{\left\|\nabla \left(\frac{1}{2}\mathbf{y}^\top \left((\tilde{\mathbf{K}} + \sigma^2 \mathbf{I})^{-1} - (\mathbf{K} + \sigma^2 \mathbf{I})^{-1}\right)\mathbf{y}\right)\right\|_2}_{T_1}$$

$$+\underbrace{\left\|\frac{1}{2}\nabla \left(\log|\mathbf{K} + \sigma^2 \mathbf{I}| - \log|\tilde{\mathbf{K}} + \sigma^2 \mathbf{I}|\right)\right\|_2}_{T_2}.$$

We will bound $T_1$ and $T_2$ separately.

**Bounding $T_1$:**

$$T_1 = \frac{1}{2}\left\|\nabla_\theta \left(\mathbf{y}^\top \left((\tilde{\mathbf{K}} + \sigma^2 \mathbf{I})^{-1} - (\mathbf{K} + \sigma^2 \mathbf{I})^{-1}\right)\mathbf{y}\right)\right\|_2$$

$$= \frac{1}{2}\sqrt{\sum_{l=1}^{p}\left(\frac{\partial}{\partial \theta_l}\mathbf{y}^\top \left((\tilde{\mathbf{K}} + \sigma^2 \mathbf{I})^{-1} - (\mathbf{K} + \sigma^2 \mathbf{I})^{-1}\right)\mathbf{y}\right)^2}$$

$$\leq \frac{1}{2}\sqrt{p}\max_{1\leq l\leq p}\sqrt{\left(\frac{\partial}{\partial \theta_l}\mathbf{y}^\top \left((\tilde{\mathbf{K}} + \sigma^2 \mathbf{I})^{-1} - (\mathbf{K} + \sigma^2 \mathbf{I})^{-1}\right)\mathbf{y}\right)^2}$$

$$= \frac{1}{2}\sqrt{p}\max_{1\leq l\leq p}\left|\frac{\partial}{\partial \theta_l}\mathbf{y}^\top \left((\tilde{\mathbf{K}} + \sigma^2 \mathbf{I})^{-1} - (\mathbf{K} + \sigma^2 \mathbf{I})^{-1}\right)\mathbf{y}\right|$$

We will then bound $\left|\frac{\partial}{\partial \theta_l}\mathbf{y}^\top \left((\tilde{\mathbf{K}} + \sigma^2 \mathbf{I})^{-1} - (\mathbf{K} + \sigma^2 \mathbf{I})^{-1}\right)\mathbf{y}\right|$. Using the following equality $\frac{\partial}{\partial \theta_l}\mathbf{X}^{-1} = -\mathbf{X}^{-1}(\frac{\partial \mathbf{X}}{\partial \theta_l})\mathbf{X}^{-1}$, we can express this derivative as a quadratic form as a difference between two quadratic forms and apply standard techniques

for bounding differences between quadratic forms.

$$\left| \frac{\partial}{\partial \theta_l} \mathbf{y}^\top \left( (\tilde{\mathbf{K}} + \sigma^2 \mathbf{I})^{-1} - (\mathbf{K} + \sigma^2 \mathbf{I})^{-1} \right) \mathbf{y} \right|$$

$$\leq \|\mathbf{y}\|_2^2 \left\| \frac{\partial}{\partial \theta_l} \left( (\tilde{\mathbf{K}} + \sigma^2 \mathbf{I})^{-1} - (\mathbf{K} + \sigma^2 \mathbf{I})^{-1} \right) \right\|_2 \quad \text{CS inequality}$$

$$= \|\mathbf{y}\|_2^2 \left\| \frac{\partial}{\partial \theta_l} \left( (\tilde{\mathbf{K}} + \sigma^2 \mathbf{I})^{-1} - (\mathbf{K} + \sigma^2 \mathbf{I})^{-1} \right) \right\|_2$$

$$= \|\mathbf{y}\|_2^2 \left\| -(\tilde{\mathbf{K}} + \sigma^2 \mathbf{I})^{-1} \left( \frac{\partial}{\partial \theta_l} \tilde{\mathbf{K}} \right) (\tilde{\mathbf{K}} + \sigma^2 \mathbf{I})^{-1} + (\mathbf{K} + \sigma^2 \mathbf{I})^{-1} \left( \frac{\partial}{\partial \theta_l} \mathbf{K} \right) (\mathbf{K} + \sigma^2 \mathbf{I})^{-1} \right\|_2$$

$$= \|\mathbf{y}\|_2^2 \left\| -(\tilde{\mathbf{K}} + \sigma^2 \mathbf{I})^{-1} \left( \frac{\partial}{\partial \theta_l} \tilde{\mathbf{K}} - \frac{\partial}{\partial \theta_l} \mathbf{K} + \frac{\partial}{\partial \theta_l} \mathbf{K} \right) (\tilde{\mathbf{K}} + \sigma^2 \mathbf{I})^{-1} \right.$$
$$\left. + (\mathbf{K} + \sigma^2 \mathbf{I})^{-1} \frac{\partial}{\partial \theta_l} \mathbf{K} (\mathbf{K} + \sigma^2 \mathbf{I})^{-1} \right\|_2$$

$$= \|\mathbf{y}\|_2^2 \left\| -(\tilde{\mathbf{K}} + \sigma^2 \mathbf{I})^{-1} \left( \frac{\partial}{\partial \theta_l} \tilde{\mathbf{K}} - \frac{\partial}{\partial \theta_l} \mathbf{K} \right) (\tilde{\mathbf{K}} + \sigma^2 \mathbf{I})^{-1} \right.$$
$$\left. -(\tilde{\mathbf{K}} + \sigma^2 \mathbf{I})^{-1} \left( \frac{\partial}{\partial \theta_l} \mathbf{K} \right) (\tilde{\mathbf{K}} + \sigma^2 \mathbf{I})^{-1} + (\mathbf{K} + \sigma^2 \mathbf{I})^{-1} \left( \frac{\partial}{\partial \theta_l} \mathbf{K} \right) (\mathbf{K} + \sigma^2 \mathbf{I})^{-1} \right\|_2$$

$$\leq \|\mathbf{y}\|_2^2 \left( \underbrace{\left\| (\tilde{\mathbf{K}} + \sigma^2 \mathbf{I})^{-1} \left( \frac{\partial}{\partial \theta_l} \tilde{\mathbf{K}} - \frac{\partial}{\partial \theta_l} \mathbf{K} \right) (\tilde{\mathbf{K}} + \sigma^2 \mathbf{I})^{-1} \right\|_2}_{(a)} \right.$$

$$+ \underbrace{\left\| \left( (\tilde{\mathbf{K}} + \sigma^2 \mathbf{I})^{-1} - (\mathbf{K} + \sigma^2 \mathbf{I})^{-1} \right) \left( \frac{\partial}{\partial \theta_l} \mathbf{K} \right) (\tilde{\mathbf{K}} + \sigma^2 \mathbf{I})^{-1} \right\|_2}_{(b)}$$

$$\left. + \underbrace{\left\| (\mathbf{K} + \sigma^2 \mathbf{I})^{-1} \left( \frac{\partial}{\partial \theta_l} \mathbf{K} \right) \left( (\tilde{\mathbf{K}} + \sigma^2 \mathbf{I})^{-1} - (\mathbf{K} + \sigma^2 \mathbf{I})^{-1} \right) \right\|_2}_{(c)} \right).$$

We now explicitly bound (a), (b), and (c).

$$(a) \leq \left\| (\tilde{\mathbf{K}} + \sigma^2 \mathbf{I})^{-1} \right\|_2 \left\| \frac{\partial}{\partial \theta_l} \tilde{\mathbf{K}} - \frac{\partial}{\partial \theta_l} \mathbf{K} \right\|_2 \left\| (\tilde{\mathbf{K}} + \sigma^2 \mathbf{I})^{-1} \right\|_2$$

$$\leq \left\| (\tilde{\mathbf{K}} + \sigma^2 \mathbf{I})^{-1} \right\|_2^2 \left\| \frac{\partial}{\partial \theta_l} \tilde{\mathbf{K}} - \frac{\partial}{\partial \theta_l} \mathbf{K} \right\|_2$$

$$\leq \frac{1}{\sigma^4} \left\| \frac{\partial}{\partial \theta_l} \mathbf{K} - \frac{\partial}{\partial \theta_l} \tilde{\mathbf{K}} \right\|_2$$

$$\leq \frac{1}{\sigma^4} \gamma'_{n,m,L,l} \quad \text{(Using Lemma 5.4)}$$

$$(b) \leq \| (\tilde{\mathbf{K}} + \sigma^2 \mathbf{I})^{-1} - (\mathbf{K} + \sigma^2 \mathbf{I})^{-1} \|_2 \left\| \frac{\partial}{\partial \theta_l} \mathbf{K} \right\|_2 \| (\tilde{\mathbf{K}} + \sigma^2 \mathbf{I})^{-1} \|_2$$

$$\leq \frac{1}{\sigma^2} \| (\tilde{\mathbf{K}} + \sigma^2 \mathbf{I})^{-1} - (\mathbf{K} + \sigma^2 \mathbf{I})^{-1} \|_2 \left\| \frac{\partial}{\partial \theta_l} \mathbf{K} \right\|_2$$

$$\leq \frac{\gamma_{n,m,L}}{\sigma^6} \left\| \frac{\partial}{\partial \theta_l} \mathbf{K} \right\|_2 \quad \text{(Using Lemma B.4)}$$

where we are able to apply the last line since $\tilde{\mathbf{K}}, \mathbf{K}$ are GP kernels and thus PSD. Then

$$(b) \leq \frac{\gamma_{n,m,L}}{\sigma^6} \left\| \frac{\partial}{\partial \theta_l} \mathbf{K} \right\|_2$$

$$\leq C_n \frac{\gamma_{n,m,L}}{\sigma^6} \text{ by assumption}$$

and finally

$$(c) \leq \|(\mathbf{K} + \sigma^2 \mathbf{I})^{-1}\|_2 \left\| \frac{\partial}{\partial \theta_l} \mathbf{K} \right\|_2 \|(\tilde{\mathbf{K}} + \sigma^2 \mathbf{I})^{-1} - (\mathbf{K} + \sigma^2 \mathbf{I})^{-1}\|_2$$

$$\leq \frac{1}{\sigma^2} \|(\tilde{\mathbf{K}} + \sigma^2 \mathbf{I})^{-1} - (\mathbf{K} + \sigma^2 \mathbf{I})^{-1}\|_2 \left\| \frac{\partial}{\partial \theta_l} \mathbf{K} \right\|_2$$

$$\leq \frac{\gamma_{n,m,L}}{\sigma^6} \left\| \frac{\partial}{\partial \theta_l} \mathbf{K} \right\|_2 \quad \text{(Using Lemma B.4)}$$

$$\leq \frac{\gamma_{n,m,L}}{\sigma^6} C_n$$

Combining these, we obtain

$$T_1 \leq \frac{1}{2\sigma^4} \|\mathbf{y}\|^2 \sqrt{p} \max_{1 \leq l \leq p} \left( \gamma'_{n,m,L,l} + \frac{2}{\sigma^2} \gamma_{n,m,L} C_n \right)$$

**Bounding the Trace Term Error ($T_2$):**

To bound the error in the trace term of the gradient, we begin with the standard identity for the gradient of a log-determinant with respect to a parameter $\theta_l$ (see e.g. (Petersen et al., 2008)):

$$\frac{\partial}{\partial \theta_l} \log |\mathbf{K}(\theta)| = \mathrm{Tr} \left( \mathbf{K}(\theta)^{-1} \frac{\partial \mathbf{K}(\theta)}{\partial \theta_l} \right)$$

The term $T_2 = \left\| \frac{1}{2} \nabla \left( \log |\mathbf{K} + \sigma^2 \mathbf{I}| - \log |\tilde{\mathbf{K}} + \sigma^2 \mathbf{I}| \right) \right\|_2$ represents the L2 norm of the gradient error vector for the log-determinant. We first analyze the magnitude of its $l$-th component, which we denote $(T_2)_l$:

$$(T_2)_l = \frac{1}{2} \left( \mathrm{Tr} \left( (\mathbf{K} + \sigma^2 \mathbf{I})^{-1} \frac{\partial \mathbf{K}}{\partial \theta_l} \right) - \mathrm{Tr} \left( (\tilde{\mathbf{K}} + \sigma^2 \mathbf{I})^{-1} \frac{\partial \tilde{\mathbf{K}}}{\partial \theta_l} \right) \right)$$

By applying the triangle inequality to the trace, we can decompose the error:

$$|(T_2)_l| \leq \frac{1}{2} \left| \mathrm{Tr} \left( ((\mathbf{K} + \sigma^2 \mathbf{I})^{-1} - (\tilde{\mathbf{K}} + \sigma^2 \mathbf{I})^{-1}) \frac{\partial \mathbf{K}}{\partial \theta_l} \right) \right| + \frac{1}{2} \left| \mathrm{Tr} \left( (\tilde{\mathbf{K}} + \sigma^2 \mathbf{I})^{-1} \left( \frac{\partial \mathbf{K}}{\partial \theta_l} - \frac{\partial \tilde{\mathbf{K}}}{\partial \theta_l} \right) \right) \right|$$

We bound each of the two resulting parts using the property $|\mathrm{Tr}(\mathbf{AB})| \leq n\|\mathbf{A}\|_2\|\mathbf{B}\|_2$. Let $\mathbf{A}_\sigma = (\mathbf{K} + \sigma^2 \mathbf{I})$ and $\tilde{\mathbf{A}}_\sigma = (\tilde{\mathbf{K}} + \sigma^2 \mathbf{I})$.

1. The first part is bounded using $\|\mathbf{A}_\sigma^{-1} - \tilde{\mathbf{A}}_\sigma^{-1}\|_2 \leq \frac{\gamma_{n,m,L}}{\sigma^4}$ from Lemma B.4:

$$\left| \mathrm{Tr} \left( (\mathbf{A}_\sigma^{-1} - \tilde{\mathbf{A}}_\sigma^{-1}) \frac{\partial \mathbf{K}}{\partial \theta_l} \right) \right| \leq n \|\mathbf{A}_\sigma^{-1} - \tilde{\mathbf{A}}_\sigma^{-1}\|_2 \left\| \frac{\partial \mathbf{K}}{\partial \theta_l} \right\|_2 \leq n \frac{\gamma_{n,m,L}}{\sigma^4} C_n$$

2. For the second part, since $\tilde{\mathbf{K}}$ is positive semi-definite by construction, we have $\|\tilde{\mathbf{A}}_\sigma^{-1}\|_2 \leq 1/\sigma^2$. This yields:

$$\left| \mathrm{Tr} \left( \tilde{\mathbf{A}}_\sigma^{-1} (\frac{\partial \mathbf{K}}{\partial \theta_l} - \frac{\partial \tilde{\mathbf{K}}}{\partial \theta_l}) \right) \right| \leq n \|\tilde{\mathbf{A}}_\sigma^{-1}\|_2 \left\| \frac{\partial \mathbf{K}}{\partial \theta_l} - \frac{\partial \tilde{\mathbf{K}}}{\partial \theta_l} \right\|_2 \leq n \frac{1}{\sigma^2} \gamma'_{n,m,L,l}$$

Combining these gives the bound for a single component: $|(T_2)_l| = O\left(\frac{n^2 \gamma_{n,m,L}}{\sigma^4} + \frac{n \gamma'_{n,m,L,l}}{\sigma^2}\right)$.

**Combining the Bounds:**

The total error is the L2 norm of the error vector, which can be bounded by $\sqrt{p}$ times the maximum component-wise error.

$$\|\nabla_\theta \mathcal{L}(\boldsymbol{\theta}) - \nabla_\theta \tilde{\mathcal{L}}(\boldsymbol{\theta})\|_2 \leq \sqrt{p} \max_{1 \leq l \leq p} \left(|(T_1)_l| + |(T_2)_l|\right)$$

We combine our bound for $(T_2)_l$ with the bound for the quadratic term error, $|(T_1)_l| = O\left(\frac{\|\mathbf{y}\|_2^2}{\sigma^6}(n\gamma_{n,m,L} + \sigma^2 \gamma'_{n,m,L,l})\right)$, to arrive at the final expression:

$$\|\nabla_\theta \mathcal{L}(\boldsymbol{\theta}) - \nabla_\theta \tilde{\mathcal{L}}(\boldsymbol{\theta})\|_2 = O\left(\sqrt{p}\left(\frac{\|\mathbf{y}\|_2^2 n \gamma_{n,m,L}}{\sigma^6} + \frac{\|\mathbf{y}\|_2^2 \gamma'_{n,m,L}}{\sigma^4} + \frac{n^2 \gamma_{n,m,L}}{\sigma^4} + \frac{n \gamma'_{n,m,L}}{\sigma^2}\right)\right)$$

Assuming $\|\mathbf{y}\|_2^2 = O(n)$ and substituting the known bounds for $\gamma$ and $\gamma'$, the dominant terms yield the simplified scaling relationship:

$$\|\nabla_\theta \mathcal{L}(\boldsymbol{\theta}) - \nabla_\theta \tilde{\mathcal{L}}(\boldsymbol{\theta})\|_2 = O\left(c^{2d} \frac{\sqrt{p} \cdot n^3}{m^{3/d}}\right)$$

$\square$

## C.2. Proofs Related to Posterior Inference

### C.2.1. PROOF OF LEMMA 5.8

**Lemma 5.8.** *(SKI Posterior Mean Error) Let $\boldsymbol{\mu}(\cdot)$ be the GP posterior mean at a set of test points $\cdot \in \mathbb{R}^{T \times d}$ and $\tilde{\boldsymbol{\mu}}(\cdot)$ be the SKI posterior mean at those points. Then the SKI posterior mean $l^2$ error is bounded by:*

$$\|\tilde{\boldsymbol{\mu}}(\cdot) - \boldsymbol{\mu}(\cdot)\|_2$$
$$\leq \frac{c^{2d}}{\sigma^2 m^{3/d}} \sqrt{n} O\left(\max(T, n) + \frac{\sqrt{Tn} M n}{\sigma^2}\right)$$

*Proof.* We start by expressing the difference between the true and SKI posterior means:

$$\left\| \mathbf{K}_{\cdot, \mathbf{X}} \left(\mathbf{K} + \sigma^2 \mathbf{I}\right)^{-1} \mathbf{y} - \tilde{\mathbf{K}}_{\cdot, \mathbf{X}} \left(\tilde{\mathbf{K}} + \sigma^2 \mathbf{I}\right)^{-1} \mathbf{y} \right\|_2$$
$$= \left\| \left(\tilde{\mathbf{K}}_{\cdot, \mathbf{X}} - \mathbf{K}_{\cdot, \mathbf{X}}\right) \left(\tilde{\mathbf{K}} + \sigma^2 \mathbf{I}\right)^{-1} \mathbf{y} + \mathbf{K}_{\cdot, \mathbf{X}} \left[\left(\tilde{\mathbf{K}} + \sigma^2 \mathbf{I}\right)^{-1} - \left(\mathbf{K} + \sigma^2 \mathbf{I}\right)^{-1}\right] \mathbf{y} \right\|_2$$

Applying the triangle inequality and submultiplicative property gives:

$$\leq \frac{1}{\sigma^2}\|\mathbf{y}\|_2\|\tilde{\mathbf{K}}_{\cdot,\mathbf{X}} - \mathbf{K}_{\cdot,\mathbf{X}}\|_2 + \|\mathbf{K}_{\cdot,\mathbf{X}}\|_2\left\|\left(\tilde{\mathbf{K}} + \sigma^2\mathbf{I}\right)^{-1} - \left(\mathbf{K} + \sigma^2\mathbf{I}\right)^{-1}\right\|_2\|\mathbf{y}\|_2$$

$$\leq \frac{\max\left(\gamma_{T,m,L}, \gamma_{n,m,L}\right)}{\sigma^2}\|\mathbf{y}\|_2 + \|\mathbf{K}_{\cdot,\mathbf{X}}\|_2\left\|\left(\tilde{\mathbf{K}} + \sigma^2\mathbf{I}\right)^{-1} - \left(\mathbf{K} + \sigma^2\mathbf{I}\right)^{-1}\right\|_2\|\mathbf{y}\|_2 \text{ Lemma } 4.4$$

$$\leq \frac{\max\left(\gamma_{T,m,L}, \gamma_{n,m,L}\right)}{\sigma^2}\|\mathbf{y}\|_2 + \sqrt{Tn}M\left\|\left(\tilde{\mathbf{K}} + \sigma^2\mathbf{I}\right)^{-1} - \left(\mathbf{K} + \sigma^2\mathbf{I}\right)^{-1}\right\|_2\|\mathbf{y}\|_2 \text{ Lemma } B.2$$

$$\leq \frac{\max\left(\gamma_{T,m,L}, \gamma_{n,m,L}\right)}{\sigma^2}\|\mathbf{y}\|_2 + \frac{\sqrt{Tn}M}{\sigma^4}\gamma_{n,m,L}\|\mathbf{y}\|_2 \text{ Lemma } B.4$$

$$= \frac{1}{\sigma^2}\|\mathbf{y}\|_2\left(\max\left(\gamma_{T,m,L}, \gamma_{n,m,L}\right) + \frac{\sqrt{Tn}M}{\sigma^2}\gamma_{n,m,L}\right)$$

$$= \frac{c^{2d}}{\sigma^2 m^{3/d}}\|\mathbf{y}\|_2 O\left(\max(T,n) + \frac{\sqrt{Tn}Mn}{\sigma^2}\right)$$

□

### C.2.2. PROOF OF LEMMA 5.9

*Proof.* First, note that

$$\|\mathbf{\Sigma}(\cdot) - \tilde{\mathbf{\Sigma}}(\cdot)\|_2 \leq \|\mathbf{K}_{\cdot,\cdot} - \tilde{\mathbf{K}}_{\cdot,\cdot}\|_2$$
$$+ \|\mathbf{K}_{\cdot,\mathbf{X}}(\mathbf{K} + \sigma^2 I)^{-1}\mathbf{K}_{\mathbf{X},\cdot} - \tilde{\mathbf{K}}_{\cdot,\mathbf{X}}(\tilde{\mathbf{K}} + \sigma^2 I)^{-1}\tilde{\mathbf{K}}_{\mathbf{X},\cdot}\|_2$$
$$\leq \gamma_{T,m,L} + \|\mathbf{K}_{\cdot,\mathbf{X}}(\mathbf{K} + \sigma^2 I)^{-1}\mathbf{K}_{\mathbf{X},\cdot} - \tilde{\mathbf{K}}_{\cdot,\mathbf{X}}(\tilde{\mathbf{K}} + \sigma^2 I)^{-1}\tilde{\mathbf{K}}_{\mathbf{X},\cdot}\|_2,$$

where we used Proposition 4.3 and the fact that $\|\mathbf{K}_{\cdot,\cdot} - \tilde{\mathbf{K}}_{\cdot,\cdot}\|_2 \leq \gamma_{T,m,L}$.

Now, we bound the second term, which is a different between two quadratic forms:

$$\|\mathbf{K}_{\cdot,\mathbf{X}}(\mathbf{K} + \sigma^2 I)^{-1}\mathbf{K}_{\mathbf{X},\cdot} - \tilde{\mathbf{K}}_{\cdot,\mathbf{X}}(\tilde{\mathbf{K}} + \sigma^2 I)^{-1}\tilde{\mathbf{K}}_{\mathbf{X},\cdot}\|_2$$
$$\leq \|\mathbf{K}_{\cdot,\mathbf{X}}(\mathbf{K} + \sigma^2 I)^{-1}\mathbf{K}_{\mathbf{X},\cdot} - \mathbf{K}_{\cdot,\mathbf{X}}(\mathbf{K} + \sigma^2 I)^{-1}\tilde{\mathbf{K}}_{\mathbf{X},\cdot}\|_2$$
$$+ \|\mathbf{K}_{\cdot,\mathbf{X}}(\mathbf{K} + \sigma^2 I)^{-1}\tilde{\mathbf{K}}_{\mathbf{X},\cdot} - \tilde{\mathbf{K}}_{\cdot,\mathbf{X}}(\tilde{\mathbf{K}} + \sigma^2 I)^{-1}\tilde{\mathbf{K}}_{\mathbf{X},\cdot}\|_2$$
$$\leq \|\mathbf{K}_{\cdot,\mathbf{X}}(\mathbf{K} + \sigma^2 I)^{-1}(\mathbf{K}_{\mathbf{X},\cdot} - \tilde{\mathbf{K}}_{\mathbf{X},\cdot})\|_2 + \|(\mathbf{K}_{\cdot,\mathbf{X}}(\mathbf{K} + \sigma^2 I)^{-1} - \tilde{\mathbf{K}}_{\cdot,\mathbf{X}}(\tilde{\mathbf{K}} + \sigma^2 I)^{-1})\tilde{\mathbf{K}}_{\mathbf{X},\cdot}\|_2$$
$$\leq \|\mathbf{K}_{\cdot,\mathbf{X}}\|_2\|(\mathbf{K} + \sigma^2 I)^{-1}\|_2\|\mathbf{K}_{\mathbf{X},\cdot} - \tilde{\mathbf{K}}_{\mathbf{X},\cdot}\|_2 + \|\mathbf{K}_{\cdot,\mathbf{X}}(\mathbf{K} + \sigma^2 I)^{-1} - \tilde{\mathbf{K}}_{\cdot,\mathbf{X}}(\tilde{\mathbf{K}} + \sigma^2 I)^{-1}\|_2\|\tilde{\mathbf{K}}_{\mathbf{X},\cdot}\|_2$$
$$\leq \frac{1}{\sigma^2}\|\mathbf{K}_{\cdot,\mathbf{X}}\|_2\|\mathbf{K}_{\mathbf{X},\cdot} - \tilde{\mathbf{K}}_{\mathbf{X},\cdot}\|_2 + \|\mathbf{K}_{\cdot,\mathbf{X}}(\mathbf{K} + \sigma^2 I)^{-1} - \tilde{\mathbf{K}}_{\cdot,\mathbf{X}}(\tilde{\mathbf{K}} + \sigma^2 I)^{-1}\|_2\|\tilde{\mathbf{K}}_{\mathbf{X},\cdot}\|_2,$$

where we used the fact that $(\mathbf{K} + \sigma^2 I)^{-1} \preceq \frac{1}{\sigma^2}I$.

Next, we bound the term $\|\mathbf{K}_{\cdot,\mathbf{X}}(\mathbf{K} + \sigma^2 I)^{-1} - \tilde{\mathbf{K}}_{\cdot,\mathbf{X}}(\tilde{\mathbf{K}} + \sigma^2 I)^{-1}\|_2$:

$$\|\mathbf{K}_{\cdot,\mathbf{X}}(\mathbf{K} + \sigma^2 I)^{-1} - \tilde{\mathbf{K}}_{\cdot,\mathbf{X}}(\tilde{\mathbf{K}} + \sigma^2 I)^{-1}\|_2$$
$$= \|\mathbf{K}_{\cdot,\mathbf{X}}(\mathbf{K} + \sigma^2 I)^{-1} - \mathbf{K}_{\cdot,\mathbf{X}}(\tilde{\mathbf{K}} + \sigma^2 I)^{-1} + \mathbf{K}_{\cdot,\mathbf{X}}(\tilde{\mathbf{K}} + \sigma^2 I)^{-1} - \tilde{\mathbf{K}}_{\cdot,\mathbf{X}}(\tilde{\mathbf{K}} + \sigma^2 I)^{-1}\|_2$$
$$\leq \|\mathbf{K}_{\cdot,\mathbf{X}}(\mathbf{K} + \sigma^2 I)^{-1} - \mathbf{K}_{\cdot,\mathbf{X}}(\tilde{\mathbf{K}} + \sigma^2 I)^{-1}\|_2 + \|\mathbf{K}_{\cdot,\mathbf{X}}(\tilde{\mathbf{K}} + \sigma^2 I)^{-1} - \tilde{\mathbf{K}}_{\cdot,\mathbf{X}}(\tilde{\mathbf{K}} + \sigma^2 I)^{-1}\|_2$$
$$= \|\mathbf{K}_{\cdot,\mathbf{X}}[(\mathbf{K} + \sigma^2 I)^{-1} - (\tilde{\mathbf{K}} + \sigma^2 I)^{-1}]\|_2 + \|(\mathbf{K}_{\cdot,\mathbf{X}} - \tilde{\mathbf{K}}_{\cdot,\mathbf{X}})(\tilde{\mathbf{K}} + \sigma^2 I)^{-1}\|_2$$
$$\leq \|\mathbf{K}_{\cdot,\mathbf{X}}\|_2\|(\mathbf{K} + \sigma^2 I)^{-1} - (\tilde{\mathbf{K}} + \sigma^2 I)^{-1}\|_2 + \|\mathbf{K}_{\cdot,\mathbf{X}} - \tilde{\mathbf{K}}_{\cdot,\mathbf{X}}\|_2\|(\tilde{\mathbf{K}} + \sigma^2 I)^{-1}\|_2$$
$$\leq \|\mathbf{K}_{\cdot,\mathbf{X}}\|_2\frac{\gamma_{n,m,L}}{\sigma^4} + \|\mathbf{K}_{\cdot,\mathbf{X}} - \tilde{\mathbf{K}}_{\cdot,\mathbf{X}}\|_2\frac{1}{\sigma^2},$$

where we used Lemma B.4 in the last inequality. Substituting this back into the main inequality, we get:

$$\|\mathbf{K}_{\cdot,\mathbf{X}}(\mathbf{K}+\sigma^2 I)^{-1}\mathbf{K}_{\mathbf{X},\cdot} - \tilde{\mathbf{K}}_{\cdot,\mathbf{X}}(\tilde{\mathbf{K}}+\sigma^2 I)^{-1}\tilde{\mathbf{K}}_{\mathbf{X},\cdot}\|_2$$

$$\leq \frac{1}{\sigma^2}\|\mathbf{K}_{\cdot,\mathbf{X}}\|_2\|\mathbf{K}_{\mathbf{X},\cdot} - \tilde{\mathbf{K}}_{\mathbf{X},\cdot}\|_2 + \left(\|\mathbf{K}_{\cdot,\mathbf{X}}\|_2\frac{\gamma_{n,m,L}}{\sigma^4} + \|\mathbf{K}_{\cdot,\mathbf{X}} - \tilde{\mathbf{K}}_{\cdot,\mathbf{X}}\|_2\frac{1}{\sigma^2}\right)\|\tilde{\mathbf{K}}_{\mathbf{X},\cdot}\|_2$$

$$= \frac{1}{\sigma^2}\|\mathbf{K}_{\cdot,\mathbf{X}}\|_2\|\mathbf{K}_{\mathbf{X},\cdot} - \tilde{\mathbf{K}}_{\mathbf{X},\cdot}\|_2 + \frac{\gamma_{n,m,L}}{\sigma^4}\|\mathbf{K}_{\cdot,\mathbf{X}}\|_2\|\tilde{\mathbf{K}}_{\mathbf{X},\cdot}\|_2 + \frac{1}{\sigma^2}\|\mathbf{K}_{\cdot,\mathbf{X}} - \tilde{\mathbf{K}}_{\cdot,\mathbf{X}}\|_2\|\tilde{\mathbf{K}}_{\mathbf{X},\cdot}\|_2.$$

Using Lemma 4.4 and the fact that $\|\mathbf{K}_{\mathbf{X},\cdot} - \tilde{\mathbf{K}}_{\mathbf{X},\cdot}\|_2 \leq \max(\gamma_{T,m,L}, \gamma_{n,m,L})$ and that $\mathbf{K}_{\cdot,\mathbf{X}} = \mathbf{K}_{\mathbf{X},\cdot}^\top$, we have $\|\mathbf{K}_{\cdot,\mathbf{X}}\|_2 = \|\mathbf{K}_{\mathbf{X},\cdot}\|_2$. Also, by assumption, $\|\mathbf{K}_{\mathbf{X},\cdot}\|_2 \leq \sqrt{Tn}M$. Using Lemma B.3, we have $\|\tilde{\mathbf{K}}_{\mathbf{X},\cdot}\|_2 \leq \sqrt{Tn}mc^{2d}M$. Substituting these bounds, we get:

$$\|\mathbf{K}_{\cdot,\mathbf{X}}(\mathbf{K}+\sigma^2 I)^{-1}\mathbf{K}_{\mathbf{X},\cdot} - \tilde{\mathbf{K}}_{\cdot,\mathbf{X}}(\tilde{\mathbf{K}}+\sigma^2 I)^{-1}\tilde{\mathbf{K}}_{\mathbf{X},\cdot}\|_2$$

$$\leq \frac{\sqrt{Tn}M}{\sigma^2}\max(\gamma_{T,m,L}, \gamma_{n,m,L}) + \frac{\gamma_{n,m,L}}{\sigma^4}(\sqrt{Tn}M)(\sqrt{Tn}mc^{2d}M) + \frac{1}{\sigma^2}\max(\gamma_{T,m,L}, \gamma_{n,m,L})(\sqrt{Tn}mc^{2d}M)$$

$$= \frac{\sqrt{Tn}M}{\sigma^2}\max(\gamma_{T,m,L}, \gamma_{n,m,L}) + \frac{\gamma_{n,m,L}}{\sigma^4}Tnmc^{2d}M^2 + \frac{\sqrt{Tn}mc^{2d}M}{\sigma^2}\max(\gamma_{T,m,L}, \gamma_{n,m,L}).$$

Finally, substituting this back into the original inequality, we obtain the desired bound:

$$\|\boldsymbol{\Sigma}(\cdot) - \tilde{\boldsymbol{\Sigma}}(\cdot)\|_2 \leq \gamma_{T,m,L} + \frac{\sqrt{Tn}M}{\sigma^2}\max(\gamma_{T,m,L}, \gamma_{n,m,L})$$

$$+ \frac{\gamma_{n,m,L}}{\sigma^4}Tnmc^{2d}M^2 + \frac{\sqrt{Tn}mc^{2d}M}{\sigma^2}\max(\gamma_{T,m,L}, \gamma_{n,m,L}).$$

$$= O\left(\frac{Tn^2mc^{4d}M^2 + \sqrt{Tn}mc^{4d}M\max(T,n)}{m^{3/d}}\right).$$

$\square$

