# OpenReview forum: "The Price of Linear Time: Error Analysis of Structured Kernel Interpolation"
_ICML.cc/2025/Conference — ICML 2025 poster_

### Official Review · Reviewer_vBju · 2025-02-14

**Overall Recommendation:** 4

**Summary:**

The authors provide a theoretical treatment of structured kernel interpolation in Gaussian processes, particularly focusing on cubic convolutional interpolation methods. They provide a comprehensive characterization of error and complexity. Their main results focus on the interplay between the number of inducing points and the error bound (both on the “elementwise” kernel values themselves and on the corresponding Gram matrix spectral norm), finding that there is a distinct difference in this relationship as the dimension increases. Namely, they show that any error tolerance can be achieved in linear time (given enough samples) for dimension less than or equal 3. Moreover, they bound the error of kernel hyper-parameter inference in SKI and posterior parameter approximation error.

## update after rebuttal: I believe my initial score holds after the rebuttal, as the paper remains a strong contribution, in my opinion.

**Claims And Evidence:**

The main claim is related to the choice of the number of inducing points to achieve (sub-)linear time complexity with respect to a fixed error tolerance. This is shown rigorously in Section 4, where it is noted that this is possible for any error tolerance for dimension at most 3, and that the error tolerance must grow in higher-dimensional settings to satisfy linear time complexity. Other key claims are related to the hyper-parameter estimation via inexact gradient ascent on the approximate log-likelihood and approximation error of the posterior parameters. In all cases, the authors leverage common techniques in the literature to prove their findings.

**Essential References Not Discussed:**

There are no missing references, to my knowledge.

**Experimental Designs Or Analyses:**

Not relevant, as there are no experiments included in this theoretical work.

**Methods And Evaluation Criteria:**

As this is a theoretical work, there are no technical evaluation criteria. The authors employ common and new methods from the literature to show their results, such as recent work related to inexact gradient descent and common techniques related to error characterization in kernel regression.

**Other Comments Or Suggestions:**

Left Column Line 125: I assume SPD kernel means positive semidefinite here and throughout the paper. I believe PSD is the more common nomenclature, and this acronym should be fully spelled out when first introduced.

Left Column Line 160-161: “compute the action of the inverse of the regularized…” I don’t think the word “action” needs to be included here. Is it not correct to just say “compute the inverse of the regularized…”

Right Column Line 116-119: The inducing points matrix U definition should not be part of the paragraph, but should be separated into its own line.

Left Column Line 216: SKI should be capitalized
Right Column Line 234: “linearly for d=2” should be “linearly for d=3”

Right Column Line 242: healthcar -> healthcare

**Other Strengths And Weaknesses:**

This is a strong paper and provides valuable theoretical insight into a widely-used method. I believe this will be a useful contribution to the community. The only weaknesses would be a bit of lack of clarity in Section 3, as the exact implementation of the SKI method (particularly, the cubic convolutional interpolation) is not clear to me.

**Questions For Authors:**

Right Column Line 125-128: It is not clear to me how the weight vectors w(x) can be obtained from the convolutional interpolation kernel u(s) of definition 3.1-3.2. Can this be made more clear, or at least can a more direct connection between Section 3.2 and 3.3 be made?

**Relation To Broader Scientific Literature:**

The authors position their work in the context of three scientific domains. (1) Within the domain of theoretical analysis of Gaussian process regression/kernel regression with approximated kernels, they have contributed an analysis of a method which has previously been rarely treated: SKI. It is noted that prior works analyzing SKI focused only on one-dimensional features and did not analyze the posterior approximation. (2) In the context of the SKI literature, this works provides a theoretical basis for the widely-used method. (3) The authors also relate to theoretical works from which they utilize results, such as the inexact gradient descent literature.

**Theoretical Claims:**

I have checked through the proofs in Appendix B, and they seem relatively straightforward.

---

> ### Author Rebuttal · Authors · 2025-03-31
>
> Thank you for your support and positive assessment of our work as a valuable contribution! We are glad that you found the proofs straightforward and the work well-positioned. Regarding your points:
> * **Clarifying the SKI Implementation (Sec 3.2 / 3.3 Connection):** We will make the connection between the general SKI formulation and cubic interpolation clearer. We will explicitly state that the weight vectors $w(x)$ in $\tilde{k}(x,x^{\prime})=w(x)^{\top}K_{U}w(x^{\prime})$ are constructed using the tensor-product cubic interpolation function $g(x)$ from Definition 3.2. In this context, $g(x)$ uses the cubic kernel $u(s)$ from Definition 3.1, and the function values being interpolated, $f(c_x+hk)$, corresponding to kernel evaluations between inducing points, $k(u_i, u_j)$, which form the matrix $K_U$. We will add cross-references to make this link explicit.
> * **Writing Suggestions:**
>     * We will define the SPD (symmetric positive definite) acronym on first use (line 125) but instead replace it with the suggested PSD (positive semi-definite) nomenclature for consistency.
>     * We will rephrase "compute the action of the inverse" to simply "compute the inverse times a vector" or similar.
>     * We will format the definition of the inducing points matrix U on its own line for clarity.
>     * We will ensure consistent capitalization of SKI.
>     * Regarding the comment on Line 234 ("linearly for $d=2$" should be "linearly for $d=3$"): Thank you for prompting us to double-check. You are correct and we will fix this. We appreciate the careful reading.
>     * We will fix the typo "healthcar".
>
> Thank you again for your encouraging review and constructive suggestions.

---

> > ### Comment · Reviewer_vBju · 2025-04-05
> >
> > Thank you for addressing my concerns. I believe my original score holds, and this is a good contribution.

---

> > > ### Author Response · Authors · 2025-04-05
> > >
> > > Thank you for your support!

---

### Official Review · Reviewer_2Sc8 · 2025-03-14

**Overall Recommendation:** 3

**Summary:**

For a fixed kernel, non-zero noise-variance parameter, and spatial dimension, the authors give an asymptotic bound on the number of inducing points needed to reach some accuracy threshold with the SKI approximation as a function of the number of data points.  Upper bounds on error in the kernel matrix, the log-likelihood, and the posterior mean and variance are all considered.  Based on their arguments, the authors recommend a scaling of inducing points like n^(d/3), at least when cubic interpolation is used for mapping onto the structured grid in SKI.

## Update after rebuttal

I do not believe that these bounds are tight; from discussion with the authors during the rebuttal period, it seems as though they are in agreement.  Nonetheless, after thinking about it, I think this is an interesting enough step in the right direction that I would be fine with seeing it in print.  I have thus updated my recommendation to a weak accept.

**Claims And Evidence:**

The authors analyze error as a function of the number of inducing points.  This is not the same as the *time* to do almost any calculation involved.  Even a matrix-vector product with SKI involves some FFTs, adding a log term into the mix.  Moreover, the SKI construction just gives a matrix-vector product; computing likelihoods and linear system solves involves some additional iterative method.  The rates of convergence can indeed be bounded by the condition number, and this is controlled via the bound on the kernel (times the number of data points) divided by the noise variance.  But these are all potentially rather pessimistic bounds with very large constants.

**Essential References Not Discussed:**

I did not see any essential references not discussed.

**Experimental Designs Or Analyses:**

There were no experiments.  I would have appreciated some, as the bounds given here are mostly upper bounds for which it is hard to judge tightness.

**Methods And Evaluation Criteria:**

This paper is purely theoretical.  No empirical evaluation is given, though I would have appreciated it in some places (it would be nice to have a sense of how tight the bounds are).

**Other Comments Or Suggestions:**

My comments all fit into the boxes above.

**Other Strengths And Weaknesses:**

Some of the proofs in the supplementary material were a bit of a slog.  The first proof, in particular, is essentially an application of distributivity; writing down an induction is fine, but seems like more than really needed.

As indicated above, I am not convinced of the tightness of most of these bounds.  Some computational experiments (or matching lower bounds, but that seems harder) would have assuaged my skepticism.

The fact that SKI suffers a curse of dimensionality is fairly well-known.  I also observe that the SKI kernel is a kernel, even if it is only an approximation to the original kernel.  However, the original kernel is usually chosen by a modeler based on a rough assessment of the regularity of the function at hand.  So altogether, it's not so straightforward to go from statements about error in the kernel matrix (for example) to statements about generalization error.

The comment about the use of LLMs in writing the model was interesting, though perhaps left me a little less trusting of the results than I otherwise would have been.

**Questions For Authors:**

No important questions.

**Relation To Broader Scientific Literature:**

The authors claim there is not much other error analysis in the literature of this type.

**Theoretical Claims:**

I checked the first proofs in the supplement, though I started skimming toward the end.  There are some erroneous statements, though I am not sure how significant they are.  For hyper-parameter estimation, the authors claim that dk/dtheta is itself a kernel; but this clearly cannot be true for any family k(x,y) = phi(norm(x-y), theta) s.t. phi(0,theta) is constant, since in that case the diagonal of the kernel is zero.  Hence, for example, this statement is invalid for most length-scale parameters for standard kernels -- the derivative matrices will be symmetric, but indefinite.

---

> ### Author Rebuttal · Authors · 2025-04-01
>
> Thank you for your thorough and critical reading of our manuscript. We appreciate the detailed feedback, particularly regarding the assumption about kernel derivatives.
>
> **Regarding the critique of Assumption 5.3:** You are absolutely correct. Our claim that the partial derivative of the kernel $k_\theta$ with respect to lengthscale $\ell$ for RBF is a valid SPD kernel (Assumption 5.3 and a short description about RBF kernel’s before the formal statement) is false. For stationary kernels $k(x,y) = \phi(||x-y||^2/\ell^2)$, the derivative with respect to $\ell$ is symmetric but indefinite. Thanks for noticing this. Crucially, we do not actually need positive definiteness. We only rely on the spectral norm error we derived: this starts with tensor-product convolutional cubic interpolation error (not for kernels specifically). It then moves to elementwise kernel error, which relies on neither positive definiteness nor symmetry. After that the spectral norm bound relies on symmetry (so that the $l_1$ and $l_\infty$ norms are equal), but not positive definiteness. We will thus adjust the assumption to not claim positive definiteness.
>
> **Regarding Bound Tightness/Experiments:** We appreciate the request for empirical validation. We performed experiments measuring $||K - \tilde{K}||_2$ on synthetic data using an RBF kernel. Specifically, we set the number of inducing points according to our theoretical scaling, $m = \lceil n^{d/3} \rceil$, for dimensions $d=2, 3, 4$. We then plotted the log of the approximation error against the log of the sample size $n$.
>
> Our theoretical bounds predict that the error should asymptotically approach a constant $O(1)$ for this scaling ($m \propto n^{d/3}$). However, the experiments revealed a more favorable trend: for all tested dimensions ($d=2, 3, 4$), the approximation error actually decreased as $n$ increased. This suggests our theoretical upper bounds are likely pessimistic as you suggested and that the method's practical scaling performance is better than the worst-case guarantee.
>
> Furthermore, a careful re-reading of our theory (Theorem 4.5 proof) confirmed the correct threshold condition is $d<3$ vs $d \ge 3$; we had made a typo and written $d \le 3$ vs $d > 3$ and will correct this throughout the paper. We will include plots and details of these experiments in the next revision of the paper.
>
> **Regarding Novelty (Curse of Dimensionality):** While the curse of dimensionality impacting SKI is known, our contribution lies in its theoretical characterization that we can control error by growing inducing points at an $n^{d/3}$ rate and that for $d<3$ this guarantees linear time controlled error, further supported by our empirical findings showing performance exceeding the pessimistic bounds.
>
> **Regarding Proof Style/LLMs:** Lemma A.1 was included for completeness. Regarding LLMs, they assisted in outlining and literature search, but all final proofs were a combination of human and AI derived and human verified.
>
> **Regarding Generalization Error:** We agree kernel matrix error isn't the full story. Our bounds on the posterior mean and covariance approximation errors (Lemmas 5.9, 5.10) provide a more direct theoretical link between the SKI approximation and the resulting predictive distribution's accuracy.
>
> Thank you again for your detailed critique. Clarifying the assumption issue and incorporating the empirical results will significantly strengthen the paper.

---

> > ### Comment · Reviewer_2Sc8 · 2025-04-04
> >
> > Thank you for your responses.  I am still on the fence about this, but warmer than I was.  The general sentiment of the reviews seems warm enough that I expect this will likely be accepted -- anticipating that this might be the case, I note a few smaller issues below that I suggest correcting in a final version.
> >
> > Regarding tightness of the bounds, I think it is worth noting that interpolation will generally work better in the "tails" of the kernel where there is increased smoothness.  So while I think the bound on the spectral norm error is basically correct, I think that it has some built-in looseness for many practical situations.
> >
> > Regarding generalization and time: I think it is worth emphasizing the assumption here that the kernel hyperparameters (including the noise variance term) will not vary as a function of n.  Particularly, the results here depend strongly on the assumption that the noise variance term is a nonzero constant.  This is not often the case one sees where scaling up the number of measurements -- often the length scale decreases, and sometimes the noise variance term decreases.
> >
> > On more minor notes:
> >
> > - The statement that big-O is only being used for the cubic case should be moved up to the start of 4.1 (before Lemma 4.2, which uses the convention but does not state it).
> >
> > - In the header to 4.1, change "Ski" to "SKI"
> >
> > - Just before Lemma 5.9, change "Similarly to for the score function error" to "Similarly for the score function error"
> >
> > - Change "healthcar" to "healthcare" in the broader impact.

---

> > > ### Author Response · Authors · 2025-04-05
> > >
> > > Thank you for the additional feedback. Regarding the fixed hyperparameters assumption (especially noise variance), you raise a valid point about the scope of our analysis. We will add a brief discussion acknowledging this assumption in the final version and noting that analyzing the setting where hyperparameters vary with $n$ is an important direction for future investigation.
> > >
> > > Your comment on bound tightness/tails also aligns well with our experimental findings showing potentially pessimistic bounds, as discussed in our rebuttal. We confirm all suggested minor edits (Big-O statement, casing, grammar, typos) will be corrected in the camera-ready version.
> > >
> > > Thanks again for the constructive feedback.

---

### Official Review · Reviewer_AYYe · 2025-03-15

**Overall Recommendation:** 4

**Summary:**

In this paper, the authors prove several bounds for quantities of interest when using the structured kernel interpolation approximation of Gaussian processes. These bounds include element-wise errors in the kernel matrix, the spectral norm of the difference between the approximated and true Gram matrices, the log marginal likelihood (referred to as the score function in the paper), and the $L^2$/spectral norm error between the posterior mean and covariance for the same set of hyperparameters. These bounds are derived using results from optimization theory and kernel ridge regression.

**Claims And Evidence:**

The authors' claims are substantiated with formal proofs provided in the appendix.

**Essential References Not Discussed:**

I believe no essential references are missing.

**Experimental Designs Or Analyses:**

There are no experiments.

**Methods And Evaluation Criteria:**

There are no evaluation criteria.

**Other Comments Or Suggestions:**

- On line 254, the theorem in question was not referenced.
- Please revise the referencing of equations and sections, as the same paragraph (lines 294–300) contains inconsistent capitalization.
- "Gram matrix" should always be capitalized, as it is derived from the surname of Jørgen Pedersen Gram. Similarly, "SKI" should always be capitalized.
- Some citations would be better suited as textual citations rather than parenthetical citations. For example: *“In the second group, the foundational work by (Wilson & Nickisch, 2015) that [..]”* would be better displayed as *“In the second group, the foundational work by Wilson & Nickisch (2015) that [..]”*. You can achieve this by replacing `\cite` with `\citet` (if using natbib) or `\textcite` (if using biblatex).
- On line 125, left column, the acronym SPD (semi-positive definite?) is not introduced. In GP literature, the acronym PSD (positive semi-definite) is more commonly used.
- Please avoid citing very old or inaccessible papers. For example, Lagrange interpolation is referenced as *“Lagrange, J.-L. Leçons élémentaires sur les mathématiques. Imprimerie de la République, 1795.”* However, it is unclear whether this citation format is correct, as I could not locate this version under this specific publisher. Additionally, it is not specified where in the text the interpolation is defined. The same issue applies to Kolmogorov’s 1940 paper, which is cited in relation to Gaussian processes.

**Other Strengths And Weaknesses:**

A strength of this paper is that the authors explore all facets of Gaussian process approximation, from the kernel to the final predictive distribution.

**Questions For Authors:**

No additional concerns.

**Relation To Broader Scientific Literature:**

The contributions of the paper provide an important theoretical lens on the properties of SKI, whose experimental validations show that its strength lies in lower-dimensional data; this property could be related to the authors' bounds, which change asymptotic behavior when $d \\geq 3$. As far as I know, such an analysis has not been done before and represents a good initial step toward analyzing methods based on local kernel interpolation.

**Theoretical Claims:**

I have checked some of the initial proofs and the ideas behind them, and they sound reasonable. However, I am not familiar with the literature on approximate gradient descent/ascent.

Overall, I found the paper overly dense. Although earlier results are used in the later theorems, the paper focuses on theoretical analysis, yet all proofs are relegated to the appendix. I have set my score to *weak accept* due to this. I believe the paper would be more valuable if the authors included at least a high-level description of their proofs in the main paper, as these are the paper’s main contributions.

---

> ### Author Rebuttal · Authors · 2025-03-31
>
> Thanks for your helpful review and for recognizing the paper's comprehensive scope. We acknowledge the concern regarding the density of the paper due to proofs being in the appendix. In the next version, we will incorporate additional proof intuitions or summaries of the key steps for our main theorems and lemmas into the main body of the paper to improve readability, as requested. Specifically, while the main text already outlines the core strategy for some results (like Theorem 4.5 relating error bounds to grid spacing and $m$, and Lemma 5.6 using quadratic form bounding techniques for the score function), we will add or enhance the intuitions for several key steps:
> * For **Lemma 4.1** (Interpolation Error), we will note the proof uses induction on dimension $d$, showing how the error bound accumulates multiplicatively based on the 1D case from Keys (1981).
> * For **Prop 4.3** (Gram Matrix Error) and **Lemma 4.4** (Cross-Kernel Error), we will clarify that the proofs rely on standard matrix norm inequalities ($\Vert A\Vert_2 \leq \Vert A\Vert_\infty$ for Prop 4.3 using symmetry, and $\Vert A\Vert_2 \le \sqrt{\Vert A\Vert_1 \Vert A\Vert_\infty}$ for Lemma 4.4) combined with the elementwise error bounds derived in Lemma 4.2.
> * For **Lemma 5.9** (Posterior Mean Error), which currently mentions the standard strategy, we will elaborate on the specific steps: using algebraic manipulation (add and subtract terms) and the triangle inequality to decompose the error $\Vert \mu - \tilde{\mu}\Vert_2$ into terms involving the error in the cross-kernel matrix ($K_{\cdot,X} - \tilde{K}_{\cdot,X}$) and the error in the inverse Gram matrix ($(K+\sigma^2I)^{-1} - (\tilde{K}+\sigma^2I)^{-1}$), then applying our previously derived bounds for these components (Lemma 4.4 and Lemma B.4).
> * For **Lemma 5.10** (Posterior Covariance Error), which currently highlights the quadratic form bounding, we will add detail on bounding the prior covariance term ($K_{\cdot,\cdot} - \tilde{K}_{\cdot,\cdot}$) using Prop 4.3, and explicitly mention how the update term's error decomposition utilizes the bounds derived in previous lemmas.
>
> We will address the formatting and writing suggestions:
> * We will carefully revise referencing for consistency (including theorems in Sec 4) and ensure consistent capitalization (e.g., Gram matrix, SKI).
> * We will use textual citations (\citet/\textcite) where appropriate for better flow.
> * We will define the SPD (symmetric positive definite) acronym upon first use (line 125), but change it as you suggested to PSD for consistency with common GP literature.
> * We will review the older citations (Lagrange, 1795; Kolmogorov, 1940). These were included to credit the original foundational works. We will check for standard modern references.
>
> Thank you for your constructive feedback, which will help us enhance the clarity and presentation of our work.

---

> > ### Comment · Reviewer_AYYe · 2025-04-04
> >
> > Thank you for the changes, I believe these will enhance the paper and better prime the readers when reading the the full proofs.

---

> > > ### Author Response · Authors · 2025-04-05
> > >
> > > We are glad that we were able to clarify the intuition: thank you for raising your score!

---

### Official Review · Reviewer_y1bt · 2025-03-17

**Overall Recommendation:** 3

**Summary:**

This paper presents a theoretical analysis for the structured kernel interpolation (SKI, Wilson and Nickisch, 2015), a popular method for scaling Gaussian processes (GPs). The paper proves the error bounds for the SKI gram matrix and examine its effect on GP hyperparameter estimation and posterior inference. Crucially, they identify the number of inducing points needed for error control is $O(n^d/3)$. When $d\leq 3$, SKI can achieve linear complexity for any error tolerance, but for $d>3$ the error must increase with sample size to maintain linear time. This analysis provides key insights into SKI's accuracy-scalability trade-offs.

## Updated after rebuttal:
The paper provides a novel theoretical contribution to SKI analysis. The biggest limitation of the current submission is the lack of practical implication. Furthermore, according to authors rebuttal, the experiment results suggest that the bound can be pessimistic and the practical performance of SKI seems to be better than what the worst-case theory indicates. Based on these points, I would suggest authors to incorporate a more comprehensive empirical validation in the next draft to bridge the theory and practice. I will adjust my score from 4 to 3.

**Claims And Evidence:**

This paper remains theoretical. The theoretical claims are rigorous. However, I'd appreciate some empirical simulation to validate the theoretical rates, in particular for the regime of $d>3$.

**Essential References Not Discussed:**

NA

**Experimental Designs Or Analyses:**

NA

**Methods And Evaluation Criteria:**

The paper does not involve proposing a new method.

**Other Comments Or Suggestions:**

- Sec 4.1 title, Ski -> SKI

- Section 7: typo, healthcar -> healthcare

**Other Strengths And Weaknesses:**

Strength: the paper is well-organized and for most part the intuition behind the theorems and proofs are clear.

Weakness:

- Writing clarity. I'd appreciate the authors make explicit references of the definitions for various symbols. For example, the K in section 5.1 (line 307 on page 6) is used without reference. It only became clear to me in Theorem 5.7 that $K$ stands for the number of iterations.

- Technical clarity. A lot of technical terms are used without clarification/definition. For example, in lemma 4.1 how is the error of tensor-product cubic convolution interpolation defined? And in lemma 4.2, what is $delta_{m,L}$ exactly? I only got a sense of these terms after going over the proofs. Also please clarify what norm you used when mentioning "error".

- Missing empirical validation. While I understand the paper's main contribution is the theory, I'd still appreciate some (toy) empirical study that confirms the theory, in particular in  the two regimes $d\leq 3$ and $d>3$ and see how their behaviors diverge. And I think highlighting the empirical behavior when $d>$ can really help practitioners understand the practical limitation of SKI.

**Questions For Authors:**

- In Lemma 4.2, should we have inequality $\leq$ instead of $=$ in the main equation?

- App. A: is lemma A.1 essentially the distributive property of multiplication over addition?

- The authors stated that the paper has largely used LLM. Can the authors clarify if any part of the proof is LLM-written? If so, which part, and if the authors have carefully examined through these parts?

**Relation To Broader Scientific Literature:**

This paper fills in the gap of theoretical analysis of SKI. SKI has been a popular scalable GP method, however, there has not been analysis in its approximation error and how that impacts the downstream GP learning and inference.

The error analysis builds on many existing theoretical tools, such as  convolutional cubic interpolation error analysis and gradient ascent analysis. But the application of these analysis to GP is still novel.

**Theoretical Claims:**

I have checked the proofs of main lemmas and theorems up to section 4 and they seem to be correct.

---

> ### Author Rebuttal · Authors · 2025-04-01
>
> Thank you for your positive feedback and insightful suggestions! We are encouraged that you found the theoretical claims rigorous and the proofs correct.
>
> We will address the clarity issues raised:
> * We will explicitly define $k=1,\ldots,K$ as the iterations when we first use it in Section 5.1.
> * We will clarify the definition of interpolation error in Lemma 4.1 (as the uniform error bound over the domain using functions) and $\delta_{m,L}$ in Lemma 4.2 (as the tensor-product convolutional cubic interpolation error). We will also ensure the specific norm (spectral, elementwise, L2) is clear whenever "error" is discussed.
> * We will correct the noted typos (e.g., "healthcar").
> * We confirm Lemma 4.2 should use an inequality ($\le$), and will correct this.
> * Regarding Lemma A.1, you’re right that it essentially shows the distributive property; we included the inductive proof for formal completeness but will add a few sentences that summarize it before the full proof.
> * Regarding LLM usage: LLMs assisted with brainstorming, literature search, and initial proof drafting. However, all final mathematical arguments, proofs, and derivations presented were significantly refined, validated, and verified by the authors.
>
> Regarding the request for empirical validation: We appreciate the suggestion and performed experiments simulating the spectral norm error $||K - \tilde{K}||_2$ for varying $n$ and $d$ using an RBF kernel on synthetic data. Specifically, we set the number of inducing points according to our theoretical scaling, $m = \lceil n^{d/3} \rceil$, for dimensions $d=2, 3, 4$. We then plotted the log of the approximation error against the log of the sample size $n$.
>
> Our theoretical bound (Prop 4.3 combined with $m \propto n^{d/3}$) predicts that the error should asymptotically approach a constant $O(1)$ independent of $n$. However, our experiments showed a more favorable outcome for all tested dimensions ($d=2, 3, 4$), the approximation error actually decreased as the sample size $n$ increased with $m$ scaled precisely as $n^{d/3}$. This suggests our theoretical upper bound, particularly the $O(n)$ factor in the numerator derived from the infinity norm bound, is likely pessimistic, and the practical performance of SKI with this scaling is better than the worst-case guarantee. We will add these figures along with experimental details to the next revision.
>
> Furthermore, upon reviewing our theoretical derivation (specifically Corollary 4.7 and its proof), there is a typo: the correct regimes are $d<3$ and $d \ge 3$, but this was incorrectly stated as $d \le 3$ vs $d > 3$. We will correct this inconsistency throughout the text. Thank you again for your valuable feedback which has helped improve our paper.

---

### Decision · Program_Chairs · 2025-05-01

**Decision:**

Accept (poster)

**Comment:**

This manuscript develops error bounds that pertain to SKI, a popular method leveraged for efficient computation with Gaussian Processes. The reviewers agree that the theoretical contributions are mostly sound and sensible. Moreover, given the broad use of SKI there is value in the contribution. The reviewers do feel that there is room for improvement in aspects of the presentation. The main limitation of the manuscript in its present form is the lack of experiments to illustrate the bounds (and their looseness or tightness); the inclusion of experiments as discussed during the rebuttal will strengthen the manuscript (even if they show there is potential for further improvement of the bounds in certain situations).